# Shadow Cones: A Generalized Framework for Partial Order Embeddings

**Tao Yu**[*], **Toni J.B. Liu**[*], **Albert Tseng, and Christopher De Sa**
Cornell University
`{ty367, jl3499, at676, cmd353}@cornell.edu`

## Abstract

Hyperbolic space has proven to be well-suited for capturing hierarchical relations in data, such as trees and directed acyclic graphs. Prior work introduced the concept of entailment cones, which uses partial orders defined by nested cones in the Poincaré ball to model hierarchies. Here, we introduce the "shadow cones" framework, a physics-inspired entailment cone construction. Specifically, we model partial orders as subset relations between shadows formed by a light source and opaque objects in hyperbolic space. The shadow cones framework generalizes entailment cones to a broad class of formulations and hyperbolic space models beyond the Poincaré ball. This results in clear advantages over existing constructions: for example, shadow cones possess better optimization properties over constructions limited to the Poincaré ball. Our experiments on datasets of various sizes and hierarchical structures show that shadow cones consistently and significantly outperform existing entailment cone constructions. These results indicate that shadow cones are an effective way to model partial orders in hyperbolic space, offering physically intuitive and novel insights about the nature of such structures.

## 1 Introduction

Modern machine learning methods favor continuous data representations, as such representations can be easily used in differentiable deep models. Yet real-world data is often discrete; for example, text is composed of characters and words, and graphs consist of nodes and edges. This discrete-continuous gap has resulted in a wide variety of embedding methods that map discrete data into continuous spaces. Popular methods such as word2vec (Mikolov et al., 2013) and GloVe (Pennington et al., 2014) approach this problem by mapping into high-dimensional Euclidean spaces, which capture complex relations between data by using many dimensions.

However, not all data can be well represented in Euclidean Space (Bronstein et al., 2016; Sala et al., 2018). Hierarchical and graphical data, such as biological phylogenetic trees and social networks, are more naturally modeled in hyperbolic space (Sarkar, 2012; Sala et al., 2018). In hyperbolic space, the volume of a ball grows exponentially for large radius, which matches the number of nodes in a tree; in contrast, this volume grows polynomially in Euclidean space (Yu & De Sa, 2019). This volume advantage has made hyperbolic space popular for embedding hierarchical data, such as done in Ganea et al. (2018); Balazevic et al. (2019); Li et al. (2022); Bai et al. (2021).

In this work, we focus on learning embeddings that capture partial orders on a set of data points $X$. In a partial order, certain pairs of points $u, v \in X$ possess entailment relations. That is, if $u \prec v$, $u$ is an ancestor of $v$. Many hierarchical structures such as directed acyclic graphs (DAGs) can be expressed as partial orders, making partial orders a popular tool to represent such structures with (Vendrov et al., 2015; Ganea et al., 2018). Furthermore, since not all pairs $u, v \in X$ need to be comparable (hence the "partial" nomer), partial orders are especially useful for graph prediction tasks, where multiple embeddings with different properties can exist for a single partial order over a set.

We propose a novel and physically intuitive partial order embedding framework, which we call the "shadow cones" framework. Shadow cones use a set of hyperbolic cones derived from shadows formed by light sources and opaque objects in hyperbolic space. Entailment relations between objects are

---

[*]Equal contributions.

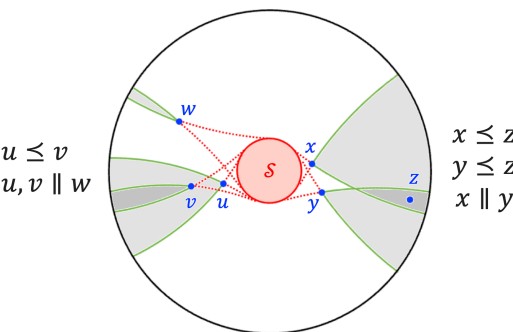

Figure 1: Example of two sets of shadow cone embeddings in the Poincaré ball, and the partial relations it encodes. Marked in black are the encoded partial relations, while in blue are the embeddings, $u, v, w$ and $x, y, z$. Marked in red are the light source ($\mathcal{S}$), and the dotted geodesics representing light rays. Shaded areas represent shadows. The symbol "$\|$" denotes negative relations between unrelated, incomparable elements.

modeled by subset relations between shadows, similar to how planets block each other out during solar eclipses. Shadow cones can be seen as a generalization of existing approaches that use hyperbolic cones to model partial orders ("entailment cones"), and are agnostic to choice of hyperbolic model. This results in better numerics and optimization properties, which allow shadow cones to empirically outperform existing entailment cone constructions on a wide variety of graph embedding tasks.

In the following sections, we present formal constructions of shadow cones in the Poincaré ball and half-plane. Our main contributions are that we:

- Introduce the *shadow cones* framework to model partial order relations, and detail two self-contained formulations in two hyperbolic models, resulting in four embedding schemes.
- Define a differentiable energy function to measure disagreement between ground truth partial orders in data and those induced by shadow cones.
- Achieve state-of-the-art results on a wide range of graph embedding tasks, lending both empirical and theoretical support to the advantages of all four embedding schemes.

## 2 PRELIMINARIES AND RELATED WORK

First, we define some key concepts in hyperbolic geometry. **Hyperbolic space** $\mathbb{H}$ is the unique simply connected Riemannian manifold with constant negative curvature $-k, k > 0$ (Anderson, 2006). There exist multiple models $\mathbb{H}$ that are isometric to each other. This work uses two such models, the Poincaré ball and the Poincaré half-space:

The **Poincaré ball** is given by $\mathcal{B}^n = \{\boldsymbol{x} \in \mathbb{R}^n : \|\boldsymbol{x}\| < 1/\sqrt{k}\}$. Distances on $\mathcal{B}^n$ are defined as

$$d_{\mathcal{B}}(\boldsymbol{x}, \boldsymbol{y}) = \frac{1}{\sqrt{k}} \operatorname{arcosh}\left(1 + 2\frac{k\|\boldsymbol{x} - \boldsymbol{y}\|^2}{(1 - k\|\boldsymbol{x}\|^2)(1 - k\|\boldsymbol{y}\|^2)}\right).$$

The **Poincaré half-space** is given by $\mathcal{U}^n = \{\boldsymbol{x} \in \mathbb{R}^n : x_n > 0\}$. Distances on $\mathcal{U}^n$ are defined as

$$d_{\mathcal{U}}(\boldsymbol{x}, \boldsymbol{y}) = \frac{1}{\sqrt{k}} \operatorname{arcosh}\left(1 + \frac{\|\boldsymbol{x} - \boldsymbol{y}\|^2}{2x_n y_n}\right)$$

**Geodesics** are Riemannian generalizations of Euclidean straight lines. They are defined as smooth curves of locally minimal length connecting two points $\boldsymbol{x}$ and $\boldsymbol{y}$ on a Riemannian manifold $\mathcal{M}$. More concepts of hyperbolic geometry are provided in Appendix A.

We now review several approaches to embedding partial orders and other hierarchical relations in Euclidean and hyperbolic space. Order embeddings were first introduced by Vendrov et al. (2015) to model partial orders in Euclidean space. Vendrov et al. (2015) defined entailment relations as subset relations between axis-parallel cones at embedded points; that is, $\boldsymbol{u} \preceq \boldsymbol{v}$ iff the cone of $\boldsymbol{v} \subseteq$ the cone

of $\boldsymbol{u}$. However, since all axis-parallel cones eventually intersect, order embeddings are incapable of model nonoverlapping categories, such as "canine" and "feline" in a taxonomy. Since volumes in Euclidean space only increase polynomially, sets of infinite cones are prone to heavy intersections. This results in provably limited space for disjoint regions that are necessary to model negative relations ($\boldsymbol{u} \parallel \boldsymbol{v}$). More recently, Zhang et al. (2022); Boratko et al. (2021) proposed box embeddings in Euclidean space using subset relations between axis-parallel boxes, to represent entailment relations. Yet these methods are subject to the drawbacks of Euclidean space, limiting their expressivity.

The "crowdedness" of Euclidean space places fundamental limits on its representation power for deep and wide hierarchical structures (Sala et al., 2018). On the other hand, volumes in hyperbolic space grow exponentially, giving clear advantages for embedding hierarchies. Many works have explored the utility of hyperbolic embeddings for hierarchical data through likelihood scoring functions (Nickel & Kiela, 2017; 2018; Yu & De Sa, 2019; Chami et al., 2020). For example Nickel & Kiela (2017) used the following heuristic composed of norms and distances to rate the likelihood of **Is-A** $\prec$ relationships: score($\textbf{Is-A}(\boldsymbol{u}, \boldsymbol{v})) = -(1 + \alpha(\|\boldsymbol{v}\| - \|\boldsymbol{u}\|))d_{\mathbb{H}}(\boldsymbol{u}, \boldsymbol{v})$. However, such approaches are limited, as they do not explicitly model partial orders and cannot easily recover learned hierarchies.

Ganea et al. (2018) introduced the concept of entailment cones, which models entailment relations by subset relations between cones rooted at points. Formally, an entailment cone at $x \in X$, $\mathfrak{S}_{\boldsymbol{x}}$, is defined by mapping a convex cone $S$ from the tangent space $T_{\boldsymbol{x}}\mathcal{M}$ to $\mathbb{H}$ using the exponential map $\mathfrak{S}_{\boldsymbol{x}} = \exp_x(S), S \subset T_{\boldsymbol{x}}\mathcal{M} = T_x\mathbb{H}$. Different choices of $S$ give different sets of entailment cones for $X$. While entailment cones offer strong empirical performance, they also suffer from initialization issues that hinder optimization. For example, the specific construction presented in Ganea et al. (2018) leaves an $\varepsilon$-hole at the origin of the Poincaré ball where cones are undefined, and care must be taken to ensure that embeddings do not land in the $\varepsilon$-hole (Ganea et al., 2018). To mitigate this issue, Ganea et al. (2018) initialize embeddings with pretrained embeddings from (Nickel & Kiela, 2017). However, this approach constrains the representational capacity of entailment cones and complicates performance analysis, as it deviates from being a self-contained framework.

Ganea's entailment cones are a special case of our "penumbral shadow cones" (section 3.2) in the Poincaré ball, and our framework provides an intuitive explanation of why this $\varepsilon$-hole exists.[1] Additionally, our shadow cones framework allows for constructions that do not suffer from the $\varepsilon$-hole problem: in Section 3.2, we will see that we can do this by instead defining penumbral cones in the Poincaré half-space and mapping the light source to infinity. Our experiments show that penumbral cones in the Poincaré half-space consistently outperform Ganea et al. (2018)'s strong baselines.

## 3 SHADOW CONES

The shadow cones framework defines entailment relations using shadows cast by a light source $\mathcal{S}$ from opaque objects representing embedded data points. The general idea is to geometrically represent the partial relation using subset relation between shadows: we say $\boldsymbol{u} \preceq \boldsymbol{v}$ when the "shadow cast" by $\boldsymbol{v}$ is a subset of the "shadow cast" by $\boldsymbol{u}$. Note that this embedding scheme automatically satisfies the transitivity of partial relations since subset relations are inherently transitive. However, explicitly characterizing the subset relations between shadowed regions is complicated. In this section, we introduce shadow cones which reduces region-region subset relations to point-region membership relations. Specifically, $\boldsymbol{v}$ is in the *shadow cone* of $\boldsymbol{u}$ iff the shadow of $\boldsymbol{v} \subseteq$ the shadow of $\boldsymbol{u}$. (Figure 1).

We categorize shadow cones into two types of cones, depending on the type of shadow used: umbral and penumbral. In umbral cones, the light source $\mathcal{S}$ is a point and data points are represented by objects with volume. Conversely, in penumbral cones, $\mathcal{S}$ has volume while data points are volumeless points. The exact shape of shadow cones depends on the shape and position of $\mathcal{S}$ and the objects associated with data points. In the following sections, we adopt $\mathcal{S}$ with shapes and positions that result in axially symmetric shadow cones. Given that hyperbolic space is endowed with continuous isometries capable of transforming non-axially symmetric placement of light sources into axially symmetric ones, our focus on symmetric cases greatly simplifies computations while at the same time do not lose any generality. We defer further discussions on isometries to Appendix D.

---

[1]Specifically, it corresponds to the interior of the "light source" inside which shadows are not cast (Remark 1).

### 3.1 UMBRAL CONES

We provide constructions for umbral cones in the Poincaré ball and half-space models. In both constructions, embedded points are centers of associated hyperbolic balls of radius $r$. In both models, these hyperbolic balls correspond to Euclidean balls with shifted centers.

**Definition 1.** *Given a point light source $\mathcal{S}$ and points $x, y \in \mathbb{H}$, we say that $y$ is in the radius-$r$ umbral cone at $x$ if for every point $u$ in the (hyperbolic) ball of radius $r$ centered at $y$, every geodesic between $u$ and $\mathcal{S}$ passes through the (hyperbolic) ball of radius $r$ centered at $x$.*

This definition formalizes the notion that the ball centered at $y$ is entirely in the shadow of the ball centered at $x$. We provide several equivalent definitions of shadow cones in Appendix C with a detailed analysis. Each resulting umbral cone is a subset of the umbral shadow, enclosed by equidistant curves (i.e., hypercycles) to the boundaries of the umbral shadow. We detail umbral cones and the boundaries in the Poincaré half-space and ball model separately below.

**Formulation 1: Umbral-half-space** We illustrate the shape of umbral cone when $\mathcal{S}$ is placed at $x_n = \infty$ in the Poincaré half-space. Light travels along vertical ray geodesics in the direction $e_n = (0, \ldots, 0, -1)$. The central axis of the cone induced by $u$ is then $A_u^{\mathcal{U}} = \{(u_1, \ldots, u_{n-1}, x_n) | 0 < x_n \leq u_n\}$. In the Poincaré half-space model, the hyperbolic ball of $u$ is a Euclidean ball with center $c_u = (u_1, \ldots, u_{n-1}, u_n \cosh \sqrt{k}r)$ and radius $r_e = u_n \sinh \sqrt{k}r$, where $-k$ is the curvature of $\mathbb{H}$. Thus, the shadow of $u$'s ball is the region directly beneath its object, or equivalently, the region within Euclidean distance $r_e$ from its central axis. Note that the umbral cone is a subset of this shadow, as *the entire ball* of $v$ needs to be in this shadow for $v \succeq u$.

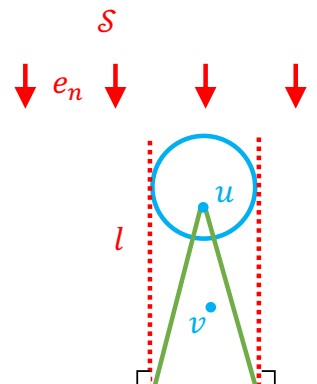

Figure 2: Umbral-half-space embeddings of partial relation $u \preceq v$. Marked in red are light source at infinity ($\mathcal{S}$), directions of light ($e_n$), and geodesic shadow boundaries ($l$). Blue is the object, and green the shadow cones.[2]

This umbral cone is better characterized by studying its boundary. Note the set of light paths tangent to the boundary of $u$'s ball is $\{(x_1, \ldots, x_{n-1}, t) | \sum_{i=1}^{n-1}(x_i - u_i)^2 = r_e^2, t > 0\}$. Let $l$ be such a light path, then one boundary of the umbral cone is the curve equidistant to $l$, and passing through $u$ (i.e., a hypercycle with axis $l$). Since $l$ is a ray-geodesic, its equidistant curve is the Euclidean straight line through $u$ and $l$'s ideal point on the $x_n = 0$–hyperplane. Thus, the umbral cone of $u$ is also a Euclidean cone with $u$ as its apex and base on the $x_n = 0$–hyperplane with Euclidean radius $r_e$. The Poincaré half-space is conformal, so the half aperture of this cone is $\theta_u = \arctan(r_e/u_n) = \arctan \sinh(\sqrt{k}r)$, which is fixed across different $u$. This allows us to test if $u \preceq v$ by simply comparing the angle between $v - u$ and $e_n$.

**Formulation 2: Umbral-Poincaré-ball** We now discuss the umbral cone when $\mathcal{S}$ is placed at the origin of the Poincaré ball model [3], which emits light along the radii. The umbral cone is symmetric with central axis $A_u^{\mathcal{B}} = \{pu | 1 \leq p < 1/(\sqrt{k}\|u\|)\}$.

Again, we characterize the umbral cone by looking at its boundary. Let $l$ be a light path tangent to the boundary of $u$'s associated ball. Assume $l$ intersects with the boundary at two ideal points $b_f, b_c$, where $b_c$ is closer to $u$ than $b_f$. The corresponding boundary of the umbral cone is the curve equidistant to $l$, and passing through $u$ (i.e., hypercycle passing through $u$ with axis $l$). This is an arc passing through three points: $b_f, b_c, u$. Thus, the related boundary of the umbral cone is the arc segment $\overline{ub_c}$ (see Figure 3).

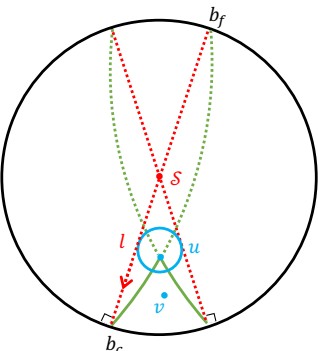

Figure 3: Umbral-Poincaré-ball embeddings of relation $u \preceq v$.

### 3.2 PENUMBRAL CONES

---

[2]Same notations and symbols apply to Figures 2-5.

[3]Again, fixing the light source at the origin causes no loss of generality, for reasons discussed in the appendix.

For penumbral cones, $\mathcal{S}$ is a convex region, while embedded points are points of no volume. We define the penumbral shadow of $\boldsymbol{u}$ as the region delineated by geodesic rays tangent to the light source's boundary $\partial\mathcal{S}$ and passing through $\boldsymbol{u}$. Consequently, $\boldsymbol{u} \preceq \boldsymbol{v}$ iff $\boldsymbol{v}$ is in the shadow of $\boldsymbol{u}$.

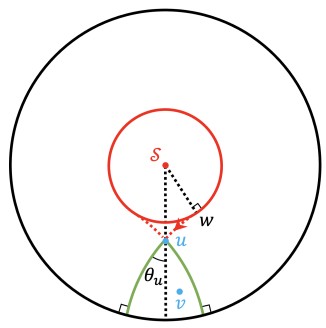

**Definition 2.** *Given a convex light source $\mathcal{S}$ and points $x, y \in \mathbb{H}$, we say that $y$ is in the* penumbral cone *at $x$ if the geodesic ray from $y$ through $x$ passes through $\mathcal{S}$.*

This definition formalizes the idea that $x$ occludes some part of the light source as seen from $y$.

Figure 4: Penumbral-Poincaré-ball embeddings of relation $u \preceq v$.

**Formulation 3: Penumbral-Poincaré-ball**  We adopt a hyperbolic ball of radius $r$ to be the light source. We parameterize the cone in the Poincaré ball model, and without loss of generality (up to an isometry), we place the center of $\mathcal{S}$ at the origin. The penumbral cone in this case is symmetric with central axis $A_{\boldsymbol{u}}^{\mathcal{B}}$.

Let $l$ be an arc geodesic tangent to the light source's boundary $\partial\mathcal{S}$ at $\boldsymbol{w}$ and passes through $\boldsymbol{u}$. Then the penumbral cone induced by $\boldsymbol{u}$ is axially-symmetric, with one boundary being the segment of $l$ starting from $\boldsymbol{u}$. The half aperture of the penumbral cone with apex $\boldsymbol{u}$ can be derived by applying the hyperbolic laws of sines (Appendix A) to the hyperbolic triangle $\triangle\boldsymbol{Swu}$, with $\angle\boldsymbol{Swu} = \pi/2$. The half aperture is then:

$$\theta_{\boldsymbol{u}} = \arcsin\left(\frac{\sinh\sqrt{k}r}{\sinh\sqrt{k}d_{\mathbb{H}}(\boldsymbol{u}, \boldsymbol{S})}\right)$$

To test the partial order of any $\boldsymbol{u}, \boldsymbol{v}$, we need to compute the angle $\phi(\boldsymbol{v}, \boldsymbol{u})$ between the cone central axis and the geodesic connecting $\boldsymbol{u}, \boldsymbol{v}$ at $\boldsymbol{u}$, i.e., $\pi - \angle\boldsymbol{Ouv}$, which is given by Ganea et al. (2018) as

$$\phi(\boldsymbol{v}, \boldsymbol{u}) = \arccos\left(\frac{\langle\boldsymbol{u}, \boldsymbol{v}\rangle(1 + k\|\boldsymbol{u}\|^2) - \|\boldsymbol{u}\|^2(1 + k\|\boldsymbol{v}\|^2)}{\|\boldsymbol{u}\|\|\boldsymbol{u} - \boldsymbol{v}\|\sqrt{1 + k^2\|\boldsymbol{u}\|^2\|\boldsymbol{v}\|^2 - 2k\langle\boldsymbol{u}, \boldsymbol{v}\rangle}}\right),$$

One can test whether $\boldsymbol{u} \preceq \boldsymbol{v}$ simply by comparing $\phi(\boldsymbol{v}, \boldsymbol{u})$ with the half aperture $\theta_{\boldsymbol{u}}$. Note that larger radius $r$ leads to wider aperture $\theta_u$, which implies larger numbers of children. In Appendix H.2, we make $r$ a learnable parameter, and observe a curious relation between the optimal radius and connectivity of the embedded dataset.

**Formulation 4: Penumbral-half-space**  The penumbral shadow cone framework is very general and do not restrict the shape of the light source. Here we introduce one more formulation with the light source shaped as **horospheres** (Izumiya, 2009), which are hyperbolic analogs of hyperplanes in Euclidean spaces. In the Poincaré half-space model, a horosphere is either a sphere tangent to the $x_n = 0$–hyperplane at an ideal point, or a Euclidean hyper-plane parallel to the $x_n = 0$–hyperplane. We use the latter as it induces symmetric shadows. In particular, we consider horospheres $\{(x_1, \ldots, x_{n-1}, \sqrt{k}e^{\sqrt{k}h})|h > 0\}$, whose boundaries $\partial\mathcal{S}$ are parallel to and with a distance $h$ from the $x_n = 0$–hyperplane.

Consider a half-circle geodesic $l$ tangent to the horosphere light source, that passes through $\boldsymbol{u}$ and ends up at infinity. Then the boundary of the penumbral cone induced by $\boldsymbol{u}$ is the segment of $l$ between $\boldsymbol{u}$ and infinity, that is, the $x_n = 0$–hyperplane. This is illustrated in Figure 5. The central axis of the cone is $A_{\boldsymbol{u}}^{\mathcal{U}}$. With some basic geometry, we derive the half aperture as $\theta_{\boldsymbol{u}} = \arcsin\left(u_n/(\sqrt{k}e^{\sqrt{k}h})\right)$.

Similarly, we compute the angle $\phi(\boldsymbol{v}, \boldsymbol{u})$ between the cone central axis and the geodesic connecting $\boldsymbol{u}, \boldsymbol{v}$ at $\boldsymbol{u}$, i.e., the angle between $\log_{\boldsymbol{u}}(\boldsymbol{v})$ and $\boldsymbol{e}_n$, where $\log$ is the logarithm map in the Poincaré half-space model, the formula of which is provided in Appendix A.

Figure 5: Penumbral-half-space embeddings of relation $u \preceq v$.

### 3.3 Some Properties of Shadow Cones

We discuss some key properties of shadow cones in this section.

Table 1: Properties of four shadow cone formulations

| Cone | Model | Form. # | Emb. Type | Convex? | Light Source | $\theta_u$ |
|------|-------|---------|-----------|---------|--------------|------------|
| Umbral | Half-Space | 1 | Ball | No | Point at $\infty$ | Fixed |
| | Poincaré Ball | 2 | Ball | No | Point at Origin | Varying |
| Penumbral | Poincaré Ball | 3 | Point | Yes | Ball at Origin | Varying |
| | Half-Space | 4 | Point | Yes | Horosphere | Varying |

**Theorem 3.1.** *The shadow cone partial orders are transitive, i.e., if $u \preceq v$ and $v \preceq w$, then $u \preceq w$.*

This theorem follows immediately from the fact that the shadow cone relation is induced by the subset relation on shadows; a rigorous proof of this theorem is provided in Appendix C. The shadow cone framework is thus well-suited framework to represent partial orders. In fact, shadow cones generalize entailment cones to a broad class of formulations, and the existing entailment cones in Ganea et al. (2018) is a special case of shadow cones, specifically, Formulation 3.

**Theorem 3.2.** *Entailment cones (Ganea et al., 2018) are equivalent to Formulation 3, penumbral cones with a hyperbolic-ball-shaped light source at the origin.*

We prove this equivalence in Appendix E. As discussed in Section 2, the $\varepsilon$-hole problem of Ganea et al. (2018)'s entailment cones is a serious limitation of the model. Here, we give an intuitive explanation of this $\varepsilon$-hole problem around the origin in the shadow cones framework.

**Remark 1** (Hole around the light source). *The shadow is not defined when the light source $\mathcal{S}$ intersects with the point $u$ (and its associated ball for umbral cones). We therefore require $d_{\mathbb{H}}(u, \mathcal{S}) > r$, which results in a hole of hyperbolic radius $r$ centered at $\mathcal{S}$.*

Umbral and penumbral cones also differ in many aspects that make them suitable for different purposes. For example,

**Theorem 3.3.** *Penumbral cones are geodesically-convex, while umbral cones are not due to the hypercycle boundary (Appendix B and C).*

The half aperture of umbral cones are fixed for $\forall u \in \mathbb{H}$, while that of penumbral cones become smaller as $u$ approaches the 0-hyperplane. We note that the hyperbolic radius $r$ and the level $h$ in penumbral cones can be trainable parameters or even a function $r(u)$ and $h(u)$ in the space. However, in our main experiments, we treat them as constants. A summary of these properties can be found in Table 1.

Apart from umbral and penumbral shadows, there are also antumbral shadows, which are formed when both the light source $\mathcal{S}$ and objects have volumes. The cones induced by antumbral shadows are mathematically equivalent to those induced by penumbral shadows, which we discuss in Appendix I.

## 4 OPTIMIZATION WITH SHADOW CONES

In this section, we use our shadow cones to design an algorithm for embedding partial relational data. Given a dataset containing partial orders between pairs of elements, we seek to learn an embedding of all elements, such that the shadow cone framework can be used to evaluate the encoded partial order, and infer missing partial orders from the dataset.

A key component in learning with shadow cones is a differentiable energy function, or loss function, which quantifies both the confidence of a correctly classified pair $(u, v)$, and the penalty associated with an incorrectly classified pair $(u, v')$. We propose to use the hyperbolic distance between $v$ and the shadow cone of $u$ as energy function.

The shortest path from $v$ to the shadow cone induced by $u$ can be classified into two distinct cases: (1) if $v$ is at a higher "altitude" than $u$, i.e. the apex of the cone, then the shortest path to the cone is the geodesic from $v$ to the apex of the cone; (2) if $v$ is at the same "altitude" as or lower than $u$, then the shortest path to the cone is the shortest geodesic from $v$ to the cone's boundary. Figure 6 illustrates the two distances for the umbral-half-space formulation.

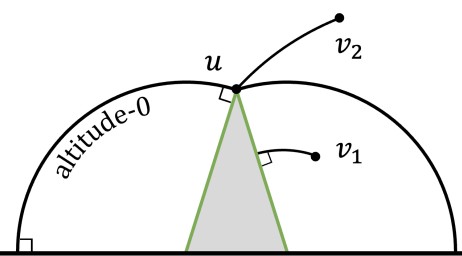

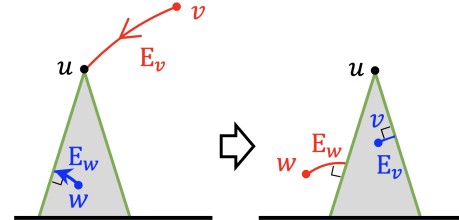

Figure 6: Shortest distances to $u$'s cone in umbral-half-space, as computed differently for negative ($v_1$) and positive ($v_2$) altitude points respectively.

Figure 7: A toy optimization example: $u \preceq v, u \parallel w$. (Left) $v$ is a child of $u$, but is wrongly initialized outside of $u$'s cone. $w$ is incomparable with $u$ but is initialized inside the cone. (Right) The energy gradients pulls $v$ inside the cone, and pushes $w$ outside. Red denotes positive energy, and blue negative.

To easily check which distance measure to use, we introduce a relative "altitude" function, $H(v, u)$. If $H(v, u) > 0$, then $u$ is at a higher altitude than the apex of $u$'s cone, corresponding to case 1. Otherwise, we are in case 2. we present this relative altitude function along with the shortest distances to the cone, using the umbral-half-space formulation. We leave the discussion of other cases, such as penumbral cones, and detailed derivations to Appendix F.

**Lemma 4.1** (Umbral-half-space). *Define temperature $t = (\sqrt{\sum_{i=1}^{n-1}(u_i - v_i)^2} - u_n \sinh \sqrt{k}r)/v_n$, then the relative altitude function of $v$ with respect to $u$ is $H(v, u) = v_n^2(1 + t^2) - u_n^2 \cosh^2 \sqrt{k}r$.*

**Theorem 4.2** (Shortest Distance to Umbral Cones). *For umbral cones with temperature $t$ and relative altitude function $H(v, u)$, the signed-distance-to-boundary function is $\frac{1}{\sqrt{k}} \operatorname{arsinh}(t) + r$. Thus, the shortest distance from $v$ to the umbral cone induced by $u$ is*

$$d(v, Cone(u)) = \begin{cases} d_{\mathbb{H}}(u, v) & \text{if } H(v, u) > 0, \\ \frac{1}{\sqrt{k}} \operatorname{arsinh}(t) + r & \text{if } H(v, u) \leq 0. \end{cases}$$

*We note that the sign of distance-to-boundary serves as another way to test partial order of $v, u$.*

To learn embeddings in a geometrically meaningful manner, we introduce "shadow loss", which is designed to draw child nodes $v$ closer to the cones of their patent nodes $u$ while simultaneously pushing away randomly sampled, negative child nodes $v'$:

$$\mathcal{L}_{\gamma_1, \gamma_2} = - \sum_{(u,v) \in P} \log \frac{\exp(-\max(E(u, v), \gamma_2))}{\sum_{(u',v') \in N} \exp(\max(\gamma_1 - E(u', v'), 0))}, \tag{1}$$

where $P$ is the edge set of positive relations, $N$ that of negative relations, and $E(u, v) = d(v, Cone(u))$ is the two-case distance defined previously. In addition, this loss allows us to choose how far to push negative samples away from the cone (distance $\gamma_1 > 0$), and how deep to pull positive samples into the cone (distance $\gamma_2 > 0$). Intuitively, positive samples shrink the embedding while negative ones dilate. In Figure 7, we show how the distance-energies are used to optimize a toy embedding for umbral-half-space.

Compared to the max-margin energy used in previous works (Vendrov et al., 2015; Ganea et al., 2018), namely $\mathcal{L} = \sum_{(u,v) \in P} E(u, v) + \sum_{(u',v') \in N} \max(0, \gamma - E(u', v'))$, our distance-based shadow loss offers several advantages: (1) The learning dynamics are refined by considering the distance or depth of both wrongly and correctly classified points relative to the cone. (2) We adopt the contrastive-style loss proposed by (Nickel & Kiela, 2017), which we demonstrate to be effective in Section 5.

Our distance-based measure offers a more nuanced understanding of hierarchical relationships compared to the angle-based energy used in (Ganea et al., 2018):$E(u, v) = \max(0, \phi(v, u) - \theta_u)$, where $\theta_u$ is the half aperture and $\phi(v, u)$ is the angle between the geodesic $uv$ and cone's central axis at $u$. This angle-based energy lacks the ability to capture the depth or confidence of the hierarchy. For example, all points $v$ on the the cone's axis associated with $u$ have $\phi(v, u) = 0$, yet some may require further optimization to better reflect hierarchical depth. Additionally, the angle-based energy, when used with max-margin energy loss in (Ganea et al., 2018), leads to vanishing gradients for misclassified negative samples in the shadow cone (Ganea et al., 2018). Our approach effectively avoids this issue.

Table 2: F1 score (%) on mammal sub-graph with best numbers **bolded**

| Non-basic-edge Percentage | Dimension = 2 | | | | | Dimension = 5 | | | | |
|---|---|---|---|---|---|---|---|---|---|---|
| | 0% | 10% | 25% | 50% | 90% | 0% | 10% | 25% | 50% | 90% |
| GBC-box | 23.4 | 25.0 | 23.7 | 43.1 | 48.2 | 35.8 | 60.1 | 66.8 | 83.8 | **97.6** |
| VBC-box | 20.1 | 26.1 | 31.0 | 33.3 | 34.7 | 30.9 | 43.1 | 58.6 | 74.9 | 69.3 |
| Entailment Cone | 54.4 | 61.0 | 71.0 | 66.5 | 73.1 | 56.3 | 81.0 | **84.1** | 83.6 | 82.9 |
| Umbral-half-space | **57.7** | 73.7 | **77.4** | **80.3** | 79.0 | **69.4** | 81.1 | 83.7 | **88.5** | 91.8 |
| Umbral-Poincaré-ball | 44.6 | 58.9 | 60.5 | 65.3 | 63.6 | 62.4 | 67.4 | 81.4 | 81.9 | 92.2 |
| Penumbral-half-space | 52.8 | **74.1** | 70.9 | 72.3 | 76.0 | 67.8 | **82.0** | 83.5 | 87.6 | 89.9 |
| Penumbral-Poincaré-ball | 44.6 | 60.8 | 62.7 | 68.4 | 67.9 | 60.8 | 69.5 | 78.2 | 84.4 | 92.6 |

Table 3: F1 score (%) on WordNet noun, MCG, and Hearst with best numbers **bolded**

| Dataset | | Noun | | | | MCG | | | | Hearst | | | |
|---|---|---|---|---|---|---|---|---|---|---|---|---|---|
| Non-basic-edge Percentage | | 0% | 10% | 25% | 50% | 0% | 10% | 25% | 50% | 0% | 1% | 2% | 5% |
| Entailment Cone | d=5 | 29.2 | 78.1 | 84.6 | 92.1 | 25.3 | 56.1 | 52.1 | 60.2 | 22.6 | 45.2 | 54.6 | 55.7 |
| | d=10 | 32.1 | 82.9 | 91.0 | 95.2 | 25.5 | 58.9 | 55.5 | 63.8 | 23.7 | 46.6 | 54.9 | 58.2 |
| Umbral-half-space | d=5 | 45.2 | 87.8 | 94.2 | 96.4 | 36.8 | 80.9 | 85.0 | 89.1 | **32.8** | 63.4 | 77.1 | 80.7 |
| | d=10 | **52.2** | **89.4** | **95.7** | **97.0** | **40.1** | **81.9** | **87.5** | **91.3** | 32.6 | **65.1** | **81.2** | **86.9** |
| Penumbral-half-space | d=5 | 44.6 | 82.6 | 86.2 | 88.3 | 35.0 | 78.6 | 81.1 | 85.3 | 26.8 | 62.8 | 72.3 | 78.8 |
| | d=10 | 51.7 | 84.1 | 88.3 | 89.8 | 37.6 | 81.9 | 85.3 | 89.2 | 28.4 | 54.4 | 68.1 | 79.3 |

## 5 EXPERIMENTS

This section showcases shadow cones' ability to represent and infer hierarchical relations on four datasets (detailed statistics in Appendix G): MCG (Wang et al., 2015; Wu et al., 2012; Li et al., 2017), Hearst patterns (Hearst, 1992; Le et al., 2019), WordNet Noun (Christiane, 1998), and its Mammal sub-graph. We consider only the **Is-A** type relations in these datasets.

**Transitive reduction and transitive closure.** MCG and Hearst are pruned until they are acyclic, as detailed in Appendix G. We then compute their transitive reduction and closure (Aho et al., 1972). Transitive reduction reduces the graph to a minimal set of relations from which all other relations could be inferred. Consistent with (Ganea et al., 2018), we refer to this minimal set as the "basic" edges. On the other hand, transitive closure encompasses all pairs of points connected via transitivity. Transitive reduction and closure respectively offer the most succinct and exhaustive representations of DAGs.

**Training and testing.** Non-basic edges can be inferred from the basic ones by transitivity, but the reverse is not true. Therefore, it is critical to include all basic edges in the training sets. Consistent with (Ganea et al., 2018), we create training sets with varying levels of difficulty by progressively adding 1% to 90% of the non-basic edges. The remaining 10% of non-basic edges are evenly divided between the validation and test sets. Our main testing regime is to first train an embedding using a collection of basic and non-basic edges, and then use it to predict unseen non-basic edges. In Appendix H we discuss other downstream tasks such as distance-based classification.

**Initialization.** The initial embeddings have been observed to be important for training (Ganea et al., 2018). Following the convention established by (Nickel & Kiela, 2017), we initialize our embeddings as a uniform distribution around origin in $[-\varepsilon, \varepsilon]$ in each dimension. Note the origin of the Poincaré half-space model is $(0, \ldots, 0, 1/\sqrt{k})$. In the Poincaré ball model, since embedded points and light sources can not overlap, we project the initialized embeddings away until they are at least of distance $r$ (radius of hyperbolic ball) from the origin. We note that (Ganea et al., 2018) adopted a pretrained 100-epoch embedding from (Nickel & Kiela, 2017) as initialization.

We benchmark against the entailment cones proposed in (Ganea et al., 2018) by following their original implementation with max-margin angle-based energy loss, which to our knowledge is state-of-the-art model using cone embeddings in hyperbolic space. For completion, we also compare with the latest box embeddings in Euclidean space: GBC-box and VBC-box (Zhang et al., 2022; Boratko et al., 2021). Table 2 summarizes all four shadow cone's performance on Mammal, which nearly outperform all previous methods, Euclidean and Hyperbolic, in terms of generalization.

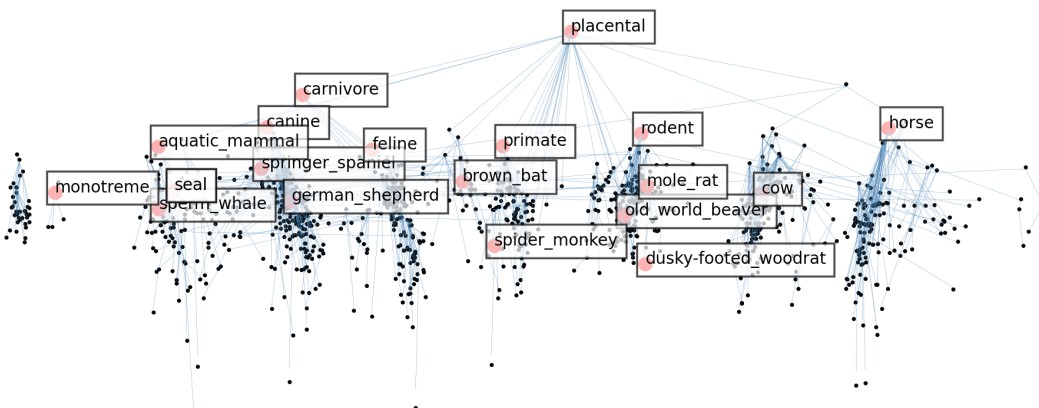

Figure 8: Mammal Sub-graph Embeddings in the 2D Umbral-Half-Space Formulation. Black points represent the embedded nodes, while blue lines the basic edges between them.

**Discussion on the impact of $\varepsilon$-hole.** We note that the Poincaré half-space formulations of shadow cones consistently outperform their Poincaré-ball counterparts. We attribute this difference to the initial embeddings prior to optimization. In the Poincaré-ball model, due to the hole around the light source at the origin, the initial embeddings are randomly projected away onto the boundary of the hole, which can destroy the hierarchical relationship between two objects by potentially pushing them to opposite directions. However, in contrast, shadow cones in the Poincaré half-space model do not suffer from this issue as the light source is placed far away from the hyperbolic origin.

On large datasets (WordNet Noun, MCG, and Hearst), we compare shadow cones in the Poincaré half-space against the baseline, as shown in Table 3. In all experiments, umbral-half-space cones consistently outperform the baselines and penumbral-half-space cones for all non-basic-edge percentages. This is likely because penumbral-half-space has a height limit while umbral-half-space does not. Finally, we visualize one of our trained embeddings: Umbral-half-space on Mammals, in Figure 8. The points represent taxonomic names, with blue edges indicating basic relations. It's noteworthy that the learnt embedding naturally organizes points into clusters, roughly corresponding to families. The depth of points within these clusters may be interpreted as taxonomic ranks, such as the Canidae family to the German Shepherd species. Our code is available on github [4].

## 6 CONCLUSION AND FUTURE WORK

Hierarchical structures pervade relational data. Effectively capturing such structures are essential for various applications as it may reveal hidden patterns and dependencies within intricate datasets. We introduce the shadow cone framework, a physically inspired approach for defining partial orders on hyperbolic space and general Riemannian manifolds. Empirical results show its superior representation and generalization capabilities, outperforming leading entailment cones. Moreover, different shadow cones may be suitable for different data structures. For example, Penumbral-half-space cones can have various apertures within the same embedding, and may be better suited to embed graphs with varying branching factors than umbral cones, which have fixed aperture.

The shadow cones framework also allows one to capture multiple relation types within one embedding by using multiple light sources, with the shadows cast by each light source capturing one type of relation (Appendix J). This approach may enable more comprehensive representation of complex relational datasets.

Although our primary focus in this work is on learning embeddings directly from partial-order datasets, the shadow cone framework is versatile enough to implicitly learn latent hierarchical structures in data. For instance, a study by Tseng et al. (2023) enhances the performance of various transformer models by incorporating a shadow-cone-based hyperbolic attention mechanism.

---

[4]https://github.com/ydtydr/ShadowCones

ACKNOWLEDGMENTS

This work was funded by Professor Christopher De Sa's NSF award IIS-2008102, NSF CAREER award 2046760 and a gift from Google.

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

## A  ADDITIONAL KEY CONCEPTS IN HYPERBOLIC GEOMETRY

The **boundary** of $\mathbb{H}$ is the set of points infinitely far away from the origin.In the Poincaré ball, the boundary $\partial\mathcal{B}^n$ is a sphere of radius $1/\sqrt{k}$ at the "edge" of the ball. In the Poincaré half-space, the boundary $\partial\mathcal{U}^n$ is the 0-hyperplane ($x_n = 0$) and the point at infinity ($x_n = \infty$)

**Hypercycles** are curves equidistant from a geodesic axis $l$. In the Poincaré ball, hypercycles of $l$ are Euclidean circular arcs that intersect the boundary at $l$'s ideal points at non-right angles. In the Poincaré half space, if $l$ is a Euclidean semicircle, then the hypercycles of $l$ are again Euclidean circular arcs intersecting the boundary at $l$'s ideal points at non-right angles. If $l$ is a vertical ray, then hypercycles are Euclidean rays that intersect $l$'s ideal point at a non-right angle.

The **tangent space** of $x$ on a manifold $\mathcal{M} = \mathbb{H}$, denoted $T_x\mathcal{M}$, is the first order vector space approximation of $\mathcal{M}$ tangent to $\mathcal{M}$ at $x$. $T_x\mathcal{M}$ is also the set of tangent vectors of all smooth paths $\in \mathcal{M}$ through $x$. Note that the Riemannian metrics defined above are metrics of the tangent space.

The notion of locally flat tangent space allows us to define local distance metrics, $g_\mathcal{M}(x)$, everywhere on a Riemannian manifold. $g_\mathcal{M}(x)$ is termed **Riemannian metric**, and it is a positive-definite quadratic form that assigns an "infinitesimal distance" to each tangent vector at a point on the manifold.

The **exponential map**, $\exp_x(v)$, maps a vector $v \in T_x\mathbb{H}$ to another point in $\mathbb{H}$ by traveling along the geodesic from $x$ in the direction of $v$. The **logarithm map** $\log_x(y)$ is its inverse.

**Hyperbolic law of sines**, like its Euclidean counterpart, relates the angles of any hyperbolic triangle to its geodesic side lengths:

$$\frac{\sin A}{\sinh a} = \frac{\sin B}{\sinh b} = \frac{\sin C}{\sinh c} \tag{2}$$

where $A$, $B$, and $C$ are the angles of a hyperbolic triangle, and $a$, $b$, and $c$ are the geodesic edges of the sides opposite to these angles.

## B  SHADOW CONES' BOUNDARY CHARACTERIZATION

In this part, we explain intuitively the boundary computation of shadow cones and the reason why Umbral cones are not geodesically convex.

Penumbral shadow, which results from partial eclipse of a light source with volume, is produced by a point object. Therefore, the boundaries of penumbral shadow are geodesics passing through $u$ while being tangent to the light source. Note that the Penumbral Shadow($v$) $\subset$ Penumbral Shadow($u$) for any $v \in$ Penumbral Shadow($u$), therefore, all possible children of $u$, i.e., penumbral cone of $u$ coincides with the penumbral shadow of $u$, with geodesics boundary described above.

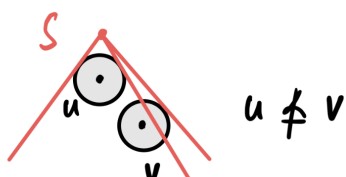

Figure 9: An example illustrating the necessity for $v$'s ball to be entirely immersed in $u$'s umbral shadow, in order that $u \preceq v$. In the marginal case shown here, $u \npreceq v$ even though the point $v$ is in $u$'s umbral shadow. This is because part of $v$'s ball and corresponding umbral shadow is outside of $u$'s umbral shadow.

On the other hand, Umbral shadow, which results from total eclipse of a point light source, is produced by an object with volume (a ball around $u$). Therefore, the boundary of umbral shadow is geodesics passing through the point light source, while being tangent to the ball around $u$. Now, let's look at all possible children $v$ of $u$, i.e., umbral cone of $u$. Since the objects $u$ and $v$ have volumes, then for $u \preceq v$ to be true, it must be that $v$ and its ball are entirely immersed in $u$'s shadow. Otherwise, the portion of $v$'s ball outside of $u$'s shadow can cast shadows outside. In Fig 9, we sketch a case where $v$'s center is in $u$'s shadow, but nevertheless $u \npreceq v$, because part of $v$'s ball is outside.

Hence, the boundary of umbral cones are the set of points equidistant from the umbral shadow, aka, hypercycles in hyperbolic space, a well-studied geometric object. In Euclidean space, the set of points equidistant from a straight line forms another, parallel straight line. In contrast, in hyperbolic space, the set of points equidistant from a geodesic do *not* necessarily form a geodesic, and is instead referred to as a hypercycle. In Fig 10, we illustrate the non-convexity of umbral cones in Poincaré

half-space. In green are the umbral cone boundaries, while in black are its convex hull - the minimum convex set - that contains the umbral cone. The boundaries of the convex hull are geodesics. We can see that the umbral cone is a strict subset of its convex hull, which means the umbral cone is not geodesically convex.

## C  PROOF OF TRANSITIVITY & GEODESIC CONVEXITY

In this section, we provide proofs to the transitivity of the partial orders induced by umbral and penumbral cones, together with the proof of penumbral cones' convexity. We first give several equivalent definitions to umbral and penumbral cones

1. $u \preceq v$ iff $v$ (and its ball) is in the shadow of $u$ (and its ball).
2. $u \preceq v$ if the shadow of $v$ (and its ball) is a subset of the shadow of $u$ (and its ball).
3. $u \preceq v$ if every geodesic between the light source $\mathcal{S}$ and $v$ (and its ball) passes through $u$ (or its ball).

Wherein, the parentheses refer to the umbral cone case. We will adopt the 3rd definition within this section.

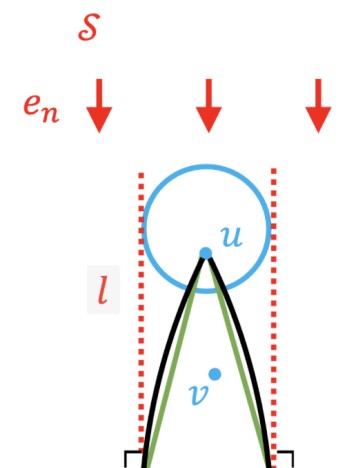

Figure 10: Umbral shadow cone (green) of $u$ in half-space model with light source $\mathcal{S}$ at infinity, where $e_n$ is the propagation direction of lights. $l$ (dotted red) denotes the boundary of umbral shadow casted by $u$ and its ball. Solid green lines denote the boundary of the umbral cone of $u$, while solid black lines delineate the geodesically convex hull of $u$'s umbral cone. This figure shows that umbral cones are not geodesically convex, because they are strict subsets of their geodesically convex hulls.

*Proof of transitivity.* We start with the umbral cone case, suppose that $x \preceq y$ and $y \preceq z$, then every geodesic between $\mathcal{S}$ and $y$'s ball passes through $x$'s ball, and every geodesic between $\mathcal{S}$ and $z$'s ball passes through $y$'s ball. Now consider any geodesic between $\mathcal{S}$ and $z$'s ball, which must passes through the $y$'s ball, but then it is also a geodesic between $\mathcal{S}$ and $y$'s ball, which must pass through $x$'s ball, that is $x \preceq z$.

For penumbral cones, suppose that $x \preceq y$ and $y \preceq z$. Consider the geodesic submanifold (isometric to the hyperbolic plane) passing through $x, y$ and $z$. Since $x \preceq y$, then the geodesic from $y$ through $x$ intersects the light source. Similarly, the geodesic from $z$ through $y$ intersects the light source. Denote these intersection points on the boundary of $\mathcal{S}$ as $a$ and $b$ respectively, then consider the geodesic ray from $z$ passing through $x$, as it passes through $z$, it either enters or exits the triangle $\triangle aby$. Since $x$ is on the line segment $ya$ now, it can't be exiting the triangle, because $y$ is on the line segment $zb$, so $z$ was already outside the triangle. Therefore, it must be entering the triangle and it must exit the triangle at some point along one of the other sides.

Note that it can't exit along the side $yb$, because $z$ is already on that line, and it can't intersect the line twice, so it must exit along the side $ab$, but that side is entirely within the light source, because $a$ and $b$ are on the light source and the light source is convex. therefore $x \preceq z$.

Worthy to mention, in the proof of penumbral cones, we used the fact that $\mathcal{S}$ is convex, so is the intersection of $\mathcal{S}$ with any geodesic submanifold of dimension 2. In fact, we only need the intersection with any geodesic submanifold of dimension 2 to be connected, but convexity of $\mathcal{S}$ suffices.  $\square$

*Proof of geodesic convexity for penumbral cones.* This can be proved following a similar pattern as last proof. Suppose that $x \succeq y$ and $x \succeq z$. Let $w$ be any point on the geodesic line segment $yz$, again we consider the geodesic submanifold passing through $x, y$ and $z$, which also passes through $w$. The geodesic from $y$ through $x$ intersects the light source, so is the geodesic from $z$ through $x$. Denote these intersection points as $a, b$ respectively. Now consider the geodesic from $w$ through $x$, which is contained between geodesics $xy$ and $xz$, so it will also be contained at the other side of them, i.e., $xa$ and $xb$. Also $x$ is one vertex of the triangle $\triangle abx$, so the geodesic $wx$ enters the triangle $\triangle abx$, then it must exit the triangle at some point along one of its sides. Since $wx$ intersects $xa$ and $xb$ at $x$, so it can't exit along $xa$ and $xb$ sides, because it can't intersect a line

twice. Therefore, it must exit along the side $\boldsymbol{ab}$, which is entirely within the light source, because the light source is convex. Therefore, $\boldsymbol{wx}$ intersects with the light source at some point, i.e., $\boldsymbol{w} \succeq \boldsymbol{x}$. $\quad\square$

## D ISOMETRY MAPPING LIGHT SOURCE TO ORIGIN

For the Poincaré ball model, when the light source of shadow cones is in the space but not at the origin, we utilize the following isometry to map the light source to the origin.

**Definition 3** (Isometry (Yu & De Sa, 2023)). *Let* $\text{Inv}(\boldsymbol{x}) = \frac{\boldsymbol{x}}{k\|\boldsymbol{x}\|^2}$*, then the map*

$$T_{\mathcal{S}}(\boldsymbol{x}) = -\mathcal{S} + (1 - \|\mathcal{S}\|^2)\,\text{Inv}(\text{Inv}(\boldsymbol{x}) - \mathcal{S})$$

*is an isometry of the Poncaré ball, which maps* $\mathcal{S}$ *to the origin* $\boldsymbol{O}$*, i.e.,* $T_{\mathcal{S}}(\mathcal{S}) = \boldsymbol{O}, T_{\mathcal{S}}^{-1}(\boldsymbol{O}) = \mathcal{S}$*.*

Since isometries preserve distance, geodesics and also equal distance curves (i.e., hypercycles), then both the umbral and penumbral cones are preserved under isometries. Therefore, when the light source $\mathcal{S}$ is not at the origin, we apply the isometry $T_{\mathcal{S}}$ to all embeddings, then the energy function can be computed accordingly as the light source at the origin case.

## E EQUIVALENCE OF PENUMBRAL-POINCARÉ-BALL AND GANEA ET AL. (2018)'S ENTAILMENT CONE CONSTRUCTION

*Proof.* Ganea et al. (2018) defines the entailment cones in the Poincaré ball model (of curvature $-1$) as

$$\{y \in \mathbb{B}^n | \phi(y, x) \leq \arcsin(K\frac{1 - \|x\|^2}{\|x\|})\},$$

where $\phi(y, x)$ is the angle between the half-line $(xy$ and $(ox$.

On the other hand, Penumbral-poincaré-ball is defined as

$$\{y \in \mathbb{B}^n | \phi(y, x) \leq \theta_x = \arcsin(\frac{\sinh\sqrt{k}r}{\sinh\sqrt{k}d_{\mathbb{H}}(x, O)})\},$$

where $-k$ is the curvature of the hyperbolic space. With

$$d_{\mathbb{H}}(x, O) = \frac{1}{\sqrt{k}}\text{arcosh}\left(1 + \frac{2k\|x\|^2}{1 - k\|x\|^2}\right),$$

we get

$$\theta_x = \arcsin(\frac{\sinh\sqrt{k}r}{\sinh\text{arcosh}\left(1 + \frac{2k\|x\|^2}{1 - k\|x\|^2}\right)}) = \arcsin(\frac{\sinh\sqrt{k}r}{2}\frac{1 - k\|x\|^2}{\sqrt{k}\|x\|}).$$

Note Ganea et al. (2018)'s entailment cones were studied in $k = 1$ case, then the penumbral-Poincaré-ball cone when $k = 1$ is

$$\{y \in \mathbb{B}^n | \phi(x, y) \leq \theta_x = \arcsin(\frac{\sinh r}{2}\frac{1 - \|x\|^2}{\|x\|})\},$$

set $K = \frac{\sinh r}{2}$ (since $r$ is a user-defined hyperparameter), the entailment cone reduces to penumbral-Poincaré-ball cone, a special case of shadow cones. $\quad\square$

## F MORE DETAILS ON SHORTEST HYPERBOLIC DISTANCE TO SHADOW CONES

### F.1 UMBRAL-HALF-SPACE CONES

In the main paper, we provide the shortest hyperbolic distance to the umbral-half-space cone, here we give a detailed derivation of the formulas. We start by giving the logarithm map in the Poincaré

half-space model (Yu & De Sa, 2021), let $\boldsymbol{v} = \log_{\boldsymbol{x}}(\boldsymbol{y})$, then

$$v_i = \frac{x_n}{y_n} \frac{s}{\sinh s} (y_i - x_i)$$

$$v_n = \frac{s}{\sinh s} (\cosh s - \frac{x_n}{y_n}) x_n,$$

where $s = \sqrt{k} d_{\mathbb{H}}(\boldsymbol{x}, \boldsymbol{y})$.

Note that the hyperbolic ball of radius $r$ centered at $\boldsymbol{u}$ in Poincaré half-space corresponds to an Euclidean ball with center $\boldsymbol{c_u} = (u_1, \ldots, u_{n-1}, u_n \cosh \sqrt{k}r)$ and radius $r_e = u_n \sinh \sqrt{k}r$, where $-k$ is the curvature of $\mathbb{H}$. Note that boundaries of the umbral cone induced by $\boldsymbol{u}$ are hypercycles with axis $ls$, where $l$ belongs to the set of light paths that are tangent to the boundary of $\boldsymbol{u}$'s ball: $\{(x_1, \ldots, x_{n-1}, t) | \sum_{i=1}^{n-1} (x_i - u_i)^2 = r_e^2, t > 0\}$. In order to derive the signed hyperbolic distance from $\boldsymbol{v}$ to the boundary of the umbral cone, it suffices to compute the signed distance of $\boldsymbol{v}$ to such a $l$ since hypercycles are equal-distance curves.

Now we derive the shortest signed hyperbolic distance from $\boldsymbol{v}$ to such an $l$. Let $\boldsymbol{w}$ be a point on $l$ such that $d_{\mathbb{H}}(\boldsymbol{v}, l) = d_{\mathbb{H}}(\boldsymbol{v}, \boldsymbol{w})$, then clearly the geodesic from $\boldsymbol{w}$ through $\boldsymbol{v}$ is orthogonal to $l$ at $\boldsymbol{w}$, i.e., $(\log_{\boldsymbol{w}}(\boldsymbol{v}))_n = 0 \implies \cosh s = w_n/v_n \implies d_{\mathbb{H}}(\boldsymbol{v}, l) = d_{\mathbb{H}}(\boldsymbol{v}, \boldsymbol{w}) = \operatorname{arcosh}(w_n/v_n)/\sqrt{k}$. Note that the geodesic from $\boldsymbol{w}$ through $\boldsymbol{v}$ is a half-circle-style geodesic with center on the 0-hyperplane. Since it's orthogonal to $l$ (a vertial line), hence the center of the geodesic is in fact $(w_1, \ldots, w_{n-1}, 0)$, then we have

$$w_n^2 = \sum_{i=1}^{n-1} (w_i - v_i)^2 + v_n^2,$$

by computing the radius to $\boldsymbol{v}$ and $\boldsymbol{w}$ respectively. Meanwhile, we have the following ratio between coordinates:

$$\frac{v_i - w_i}{v_i - u_i} = 1 - \frac{r_e}{\sqrt{\sum_{i=1}^{n-1} (v_i - u_i)^2}}.$$

From both equations we can derive that

$$w_n^2 = v_n^2 + (1 - \frac{r_e}{\sqrt{\sum_{i=1}^{n-1} (v_i - u_i)^2}})^2 \sum_{i=1}^{n-1} (v_i - u_i)^2$$

$$= v_n^2 + (\sqrt{\sum_{i=1}^{n-1} (v_i - u_i)^2} - r_e)^2.$$

Hence, the signed shortest distance from $\boldsymbol{v}$ to the boundary $\operatorname{Cone}(\boldsymbol{u})$ is

$$r + \frac{1}{\sqrt{k}} \operatorname{arcosh}(w_n/v_n) = r + \frac{1}{\sqrt{k}} \operatorname{arcosh}(\sqrt{1 + (\sqrt{\sum_{i=1}^{n-1} (v_i - u_i)^2} - r_e)^2/v_n^2})$$

Note that $\operatorname{arcosh}(\sqrt{1 + t^2}) = \operatorname{arsinh}(t)$, $\forall t \geq 0$, where the latter is a desired signed distance, then let

$$t = \left(\sqrt{\sum_{i=1}^{n-1} (u_i - v_i)^2} - u_n \sinh \sqrt{k}r\right) / v_n$$

be a temperature function, we derive the signed shortest distance from $\boldsymbol{v}$ to the boundary $\operatorname{Cone}(\boldsymbol{u})$ is

$$r + \frac{1}{\sqrt{k}} \operatorname{arsinh}(t).$$

In order to derive the relative altitude function of $\boldsymbol{v}$ respect to $\boldsymbol{u}$, consider when the shortest geodesic from $\boldsymbol{v}$ to the boundary of $\operatorname{Cone}(\boldsymbol{u})$ is attained at $\boldsymbol{u}$, which will be orthogonal to the boundary at $\boldsymbol{u}$, using the property of half-circle-stlye geodesic, we have that

$$v_n^2 + (\sqrt{\sum_{i=1}^{n-1} (u_i - v_i)^2} - r_e)^2 = r_e^2 + u_n^2 = u_n^2 (1 + \sinh^2 \sqrt{k}r) = u_n^2 \cosh^2 \sqrt{k}r,$$

that is, $v_n^2(1 + t^2) = u_n^2 \cosh^2 \sqrt{k}r$, hence, a natural choice of the relative altitude function is $H(\boldsymbol{v}, \boldsymbol{u}) = v_n^2(1 + t^2) - u_n^2 \cosh^2 \sqrt{k}r$. Then the shortest signed distance from $\boldsymbol{v}$ to $\text{Cone}(\boldsymbol{u})$ is $d_{\mathbb{H}}(\boldsymbol{u}, \boldsymbol{v})$ when $H(\boldsymbol{v}, \boldsymbol{u}) > 0$ and $r + \text{arsinh}(t)/\sqrt{k}$ when $H(\boldsymbol{v}, \boldsymbol{u}) \leq 0$.

### F.2 UMBRAL-POINCARÉ-BALL CONES

Similarly for umbral-Poincaré-ball cone, we have

**Lemma F.1** (Umbral-Poincaré-ball). *Denote $\alpha$ as the angle between $\boldsymbol{u}, \boldsymbol{v}$, and $\beta$ as the maximum angle spanned by the hyperbolic ball of radius $r$ associated with $\boldsymbol{u}$, then*

$$\alpha = \arccos \frac{\boldsymbol{u}^\mathsf{T} \boldsymbol{v}}{\|\boldsymbol{u}\| \|\boldsymbol{v}\|}, \quad \beta = \arcsin \frac{r}{\sinh\left(\sqrt{k} d_{\mathbb{H}}(\boldsymbol{u}, \boldsymbol{O})\right)} = \arcsin \frac{2\sqrt{k}r \|\boldsymbol{u}\|}{1 - k \|\boldsymbol{u}\|^2}.$$

*Set the temperature as*

$$t = \sinh\left(\sqrt{k} d_{\mathbb{H}}(\boldsymbol{v}, \boldsymbol{O})\right) \sin(\alpha - \beta) = \frac{2\sqrt{k} \|\boldsymbol{v}\|}{1 - k \|\boldsymbol{v}\|^2} \sin(\alpha - \beta),$$

*then the relative altitude function of $\boldsymbol{v}$ with respect to $\boldsymbol{u}$ is*

$$H(\boldsymbol{v}, \boldsymbol{u}) = \frac{1}{\sqrt{k}} \text{arcosh}\left(\frac{\cosh\left(\sqrt{k} d_{\mathbb{H}}(\boldsymbol{v}, \boldsymbol{O})\right)}{\sqrt{1 + t^2}}\right) - \frac{1}{\sqrt{k}} \text{arcosh}\left(\cosh\left(\sqrt{k} d_{\mathbb{H}}(\boldsymbol{u}, \boldsymbol{O})\right) \cosh\left(\sqrt{k}r\right)\right),$$

$$= \frac{1}{\sqrt{k}} \text{arcosh}\left(\frac{1}{\sqrt{1 + t^2}} \frac{1 + k \|\boldsymbol{v}\|^2}{1 - k \|\boldsymbol{v}\|^2}\right) - \frac{1}{\sqrt{k}} \text{arcosh}\left(\cosh\left(\sqrt{k}r\right) \frac{1 + k \|\boldsymbol{u}\|^2}{1 - k \|\boldsymbol{u}\|^2}\right).$$

In fact, boundaries of the umbral cone induced by $\boldsymbol{u}$ are hypercycles with axis $l$s, where $l$ belongs to the set of light paths that are tangent to the boundary of $\boldsymbol{u}$'s ball. We compute the signed distance of $\boldsymbol{v}$ to such a $l$ in the Poincaré ball model, which is easier since the line $\boldsymbol{O}\boldsymbol{v}$ and $l$ are geodesics, where hyperbolic laws of sines can be applied.

Denote $\alpha$ as the angle between $\boldsymbol{u}, \boldsymbol{v}$, and $\beta$ as the maximum angle spanned by the hyperbolic ball of radius $r$ associated with $\boldsymbol{u}$, then

$$\alpha = \arccos \frac{\boldsymbol{u}^\mathsf{T} \boldsymbol{v}}{\|\boldsymbol{u}\| \|\boldsymbol{v}\|}, \quad \beta = \arcsin \frac{r}{\sinh\left(\sqrt{k} d_{\mathbb{H}}(\boldsymbol{u}, \boldsymbol{O})\right)} = \arcsin \frac{2\sqrt{k}r \|\boldsymbol{u}\|}{1 - k \|\boldsymbol{u}\|^2}.$$

where the equation of $\beta$ is a result of hyperbolic laws of sines. Again using the hyperbolic laws of sines, we derive the signed distance from $\boldsymbol{v}$ to $l$ as

$$d_{\mathbb{H}}(\boldsymbol{v}, l) = \frac{1}{\sqrt{k}} \text{arsinh}\left(\sinh\left(\sqrt{k} d_{\mathbb{H}}(\boldsymbol{v}, \boldsymbol{O})\right) \sin(\alpha - \beta)\right),$$

therefore, we set the temperature as

$$t = \sinh\left(\sqrt{k} d_{\mathbb{H}}(\boldsymbol{v}, \boldsymbol{O})\right) \sin(\alpha - \beta) = \frac{2\sqrt{k} \|\boldsymbol{v}\|}{1 - k \|\boldsymbol{v}\|^2} \sin(\alpha - \beta),$$

then the signed shortest distance to the boundary of $\text{Cone}(\boldsymbol{u})$ is

$$\frac{1}{\sqrt{k}} \text{arsinh}(t) + r.$$

In order to derive the relative altitude function, we consider the altitude of $\boldsymbol{v}$, which is simply the projection of $\boldsymbol{v}$ to $l$, using hyperbolic laws of cosines, we have

$$\cosh\left(\sqrt{k} d_{\mathbb{H}}(\boldsymbol{v}, \boldsymbol{O})\right) = \cosh\left(\sqrt{k} d_{\mathbb{H}}(\boldsymbol{v}, l)\right) \cosh\left(\sqrt{k} H(\boldsymbol{v})\right),$$

Similarly for the altitude of $\boldsymbol{u}$,

$$\cosh\left(\sqrt{k} d_{\mathbb{H}}(\boldsymbol{u}, \boldsymbol{O})\right) = \cosh\left(\sqrt{k} d_{\mathbb{H}}(\boldsymbol{u}, l)\right) \cosh\left(\sqrt{k} H(\boldsymbol{u})\right) = \cosh\left(\sqrt{k}r\right) \cosh\left(\sqrt{k} H(\boldsymbol{u})\right),$$

Table 4: Dataset Statistics

|  | **Mammal** | **Noun** | **MCG** | **Hearst** |
|---|---|---|---|---|
| **Depth** | 8 | 18 | 31 | 56 |
| **# of Nodes** | 1,179 | 82,114 | 22,665 | 35,545 |
| **# of Components** | 4 | 2 | 657 | 2,133 |
| **# of Basic Relations** | 1,176 | 84,363 | 38,288 | 42,423 |
| **# of All Relations** | 5,361 | 661,127 | 1,134,348 | 6,846,245 |

combining them togeterm, the relative altitude function $H(\boldsymbol{v}, \boldsymbol{u}) = H(\boldsymbol{v}) - H(\boldsymbol{u})$ is

$$H(\boldsymbol{v}, \boldsymbol{u}) = \frac{1}{\sqrt{k}} \operatorname{arcosh}\left(\frac{\cosh\left(\sqrt{k}d_{\mathbb{H}}(\boldsymbol{v}, \boldsymbol{O})\right)}{\sqrt{1+t^2}}\right) - \frac{1}{\sqrt{k}} \operatorname{arcosh}\left(\cosh\left(\sqrt{k}d_{\mathbb{H}}(\boldsymbol{u}, \boldsymbol{O})\right)\cosh\left(\sqrt{k}r\right)\right),$$

$$= \frac{1}{\sqrt{k}} \operatorname{arcosh}\left(\frac{1}{\sqrt{1+t^2}}\frac{1+k\|\boldsymbol{v}\|^2}{1-k\|\boldsymbol{v}\|^2}\right) - \frac{1}{\sqrt{k}} \operatorname{arcosh}\left(\cosh\left(\sqrt{k}r\right)\frac{1+k\|\boldsymbol{u}\|^2}{1-k\|\boldsymbol{u}\|^2}\right).$$

Then the shortest signed distance from $\boldsymbol{v}$ to $\text{Cone}(\boldsymbol{u})$ is $d_{\mathbb{H}}(\boldsymbol{u}, \boldsymbol{v})$ when $H(\boldsymbol{v}, \boldsymbol{u}) > 0$ and $r + \operatorname{arsinh}(t)/\sqrt{k}$ when $H(\boldsymbol{v}, \boldsymbol{u}) \leq 0$.

### F.3 PENUMBRAL CONES

For penumbral cones, things are simpler since the boundary of penumbral cones are geodesics, where we can freely apply the hyperbolic laws of sines to hyperbolic triangles.

**Theorem F.2** (Shortest Distance to Penumbral Cones). *For penumbral cones, the temperature is defined as $t = \phi(\boldsymbol{v}, \boldsymbol{u}) - \theta_{\boldsymbol{u}}$, where $\phi(\boldsymbol{v}, \boldsymbol{u})$ is the angle between the cone central axis and the geodesic connecting $\boldsymbol{u}, \boldsymbol{v}$ at $\boldsymbol{u}$, $\theta_{\boldsymbol{u}}$ is half-aperture of the cone. The relative altitude function is $H(\boldsymbol{v}, \boldsymbol{u}) = t - \pi/2$, and the shortest distance from $\boldsymbol{v}$ to the penumbral cone induced by $\boldsymbol{u}$ is*

$$d(\boldsymbol{v}, Cone(\boldsymbol{u})) = \begin{cases} d_{\mathbb{H}}(\boldsymbol{u}, \boldsymbol{v}) & \text{if } H(\boldsymbol{v}, \boldsymbol{u}) > 0, \\ \frac{1}{\sqrt{k}} \operatorname{arsinh}\left(\sinh\left(\sqrt{k}d_{\mathbb{H}}(\boldsymbol{u}, \boldsymbol{v})\right)\sin t\right) & \text{if } H(\boldsymbol{v}, \boldsymbol{u}) \leq 0, \end{cases}$$

*where the second formula represents the signed-distance-to-boundary.*

We derive the relative altitude function first, note that $H(\boldsymbol{v}, \boldsymbol{u}) = 0$ when the geodesic from $\boldsymbol{v}$ through $\boldsymbol{u}$ is orthogonal to one boundary of the cone at $\boldsymbol{u}$, with simple geometry, the relative altitude function is $H(\boldsymbol{v}, \boldsymbol{u}) = t - \pi/2$. We derive the signed-distance-to-boundary $d_{\mathbb{H}}(\boldsymbol{v}, l)$ by simply applying the hyperbolic laws of sines:

$$\frac{\sinh\sqrt{k}d_{\mathbb{H}}(\boldsymbol{u}, \boldsymbol{v})}{\sin(\pi/2)} = \frac{\sinh\sqrt{k}d_{\mathbb{H}}(\boldsymbol{v}, l)}{\sin t},$$

then we get that $d_{\mathbb{H}}(\boldsymbol{v}, l) = \frac{1}{\sqrt{k}} \operatorname{arsinh}\left(\sinh\left(\sqrt{k}d_{\mathbb{H}}(\boldsymbol{u}, \boldsymbol{v})\right)\sin t\right)$. In summary, the shortest distance from $\boldsymbol{v}$ to the penumbral cone induced by $\boldsymbol{u}$ is

$$d(\boldsymbol{v}, \text{Cone}(\boldsymbol{u})) = \begin{cases} d_{\mathbb{H}}(\boldsymbol{u}, \boldsymbol{v}) & \text{if } H(\boldsymbol{v}, \boldsymbol{u}) > 0, \\ \frac{1}{\sqrt{k}} \operatorname{arsinh}\left(\sinh\left(\sqrt{k}d_{\mathbb{H}}(\boldsymbol{u}, \boldsymbol{v})\right)\sin t\right) & \text{if } H(\boldsymbol{v}, \boldsymbol{u}) \leq 0, \end{cases}$$

where the second formula represents the signed-distance-to-boundary.

## G  TRAINING DETAILS

**Data pre-processing and statistics.**  WordNet data are directly taken from (Ganea et al., 2018), which was already a DAG. MCG and Hearst are taken from (Wang et al., 2015; Wu et al., 2012) and (Hearst, 1992) respectively. Since these datasets are orders of magnitudes larger than WordNet, we take the $50,000$ relations with the highest confidence score. We note that there are numerous cycles

even in the truncated MCG and Hearst graphs. To obtain DAGs from these graphs, we randomly remove 1 relation from a detected cycle until no more cycles are found. The resulting four DAGs have vastly different hierarchy structures. To roughly characterize these structures, we use longest path length as a proxy for the depth of DAG and number of components (disconnected sub-graphs) as the width.

While the WordNet Noun and MCG datasets are trained with a maximum of 50% non-basic edges, we limit Hearst training to 5%. This is due to Hearst's transitive closure being more than $10\times$ larger than that of WordNet Noun, despite having comparable numbers of basic relations. We note that a data set's complexity is better reflected by the size of its basic edges than that of the transitive closure, as the latter scales quadratically with depth. For instance, although Hearst possesses only half as many basic relations as Noun, its transitive closure is tenfold larger. This can be attributed to its depth - 56 compared to Noun's 18 and MCG's 31. Full data statistics can be found in Table 4.

**Negative sampling.**    Negative samples for testing: For each positive pair in the transitive closure $(u, v)$, we create 10 negative pairs by randomly selecting 5 nodes $v'$, and 5 nodes $u'$ such that the corrupted pairs $(u, v')$ and $(u', v)$ are not present in the transitive closure. As a result, this 'true negative set' is ten times the size of the transitive closure. We choose negative set size equals 10 as a result of following (Ganea et al., 2018).

Negative samples for training are generated in a similar fashion: For each positive pair $(u, v)$, randomly corrupt its edges to form 10 negative pairs $(u, v')$ and $(u', v)$, while ensuring that these negative pairs do not appear in the training set. We remark that since the training set is not the full transitive closure, these dynamically generated negative pairs are impure as they might include non-basic edges.

**Burnin.**    Following (Nickel & Kiela, 2017; 2018), we adopt a burnin stage for 20 epochs at the beginning of training for better initialization, where a smaller learning rate (burnin-multiplier $0.01\times$) is used. After the burnin stage, the specified learning rate is then used ($1\times$).

**Hyper-parameters and Optimization.**    On WordNet Noun and Mammal, we train shadow cones and entailment cones for 400 epochs, following (Ganea et al., 2018). On MCG and Hearst, we train shadow cones and entailment cones for 500 epochs since due to their increased hierarchy depth. A training batchsize of 16 is used for all datasets and models. We used the standard hyperbolic space of curvature $-k$, where $k = 1$ consistently for all experiments reported in the paper, though a general $k$ value can also be used. For the margin parameters in shadow loss, we use $\gamma_2 = 0$ consistently for all experiments. We tune $\gamma_1$ and the learning rate in $\{0.01, 0.001, 0.0001\}$. For umbral cones, we tune the source radius $r$ in $\{0.01, 0.05, 0.1, 0.2, 0.3\}$, empirically $r = 0.05$ during training gives the optimal performance when evaluated under a slightly larger radius $r = 0.1$. For penumbral-half-space cones, we tune the exponentiated height $\sqrt{k}e^{\sqrt{k}h}$ in $\{2, 5, 10, 20\}$, where empirically 20 during training gives the optimal performance, which validates our assumption that the height of penumbral-half-space cones can limit its performance. Note that for shadow cones, when $H(\boldsymbol{v}, \boldsymbol{u}) > 0$, the shortest distance to the cone is $d_{\mathbb{H}}(\boldsymbol{u}, \boldsymbol{v})$, which will only pull $\boldsymbol{v}$ to be close to $\boldsymbol{u}$, apex of the shadow cone, but not pull $\boldsymbol{v}$ into the shadow cone. To solve this issue, we use $d_{\mathbb{H}}(\boldsymbol{u}', \boldsymbol{v})$ when $H(\boldsymbol{v}, \boldsymbol{u}) > 0$, where $\boldsymbol{u}'$ is derived by pushing $\boldsymbol{u}$ into its shadow cone along the central axis for a distance $\gamma_3$. We set $\gamma_3 = 0.0001$ consistently for all shadow cones. We use HTorch (Yu et al., 2023) for optimization in various models of hyperbolic space. We use RiemannianSGD for Poincaré half-space model, and RiemannianAdam for Poincaré ball model due to the ill-conditioned initialization results from the hole around the origin, where RiemannianAdam offers a much faster convergence than RiemannianSGD.

**Ease of Training.**    Our training routine is less complicated than that of the Entailment Cone (Ganea et al., 2018): 1) While the entailment cone uses pretrained 100-epoch embedding as initialization in order to avoid the $\varepsilon$-hole issue in the Poincaré-ball, we simply use random initialization around the origin as a one-stage approach. 2) Our half-space based cones and embeddings have no $\varepsilon$-hole issue at the origin, and need no extra steps to project the "stray" points out of the hole.

# H ADDITIONAL EXPERIMENTS

## H.1 DOWNSTREAM TASKS

One promising application is proposed in Tseng et al. (2023), a shadow-cone based hyperbolic attention mechanism, which can be used as a drop-in replacement for dot-product attention in transformers. Cone attention associates two points by the depth of their lowest common ancestor in a hierarchy defined by hyperbolic shadow cones. Tseng et al. (2023) tested cone attention on various models and tasks, and demonstrated how shadow cone-based attention could improve task-level performance over dot product attention and other baselines.

We would also like to supplement with a simpler experiment that demonstrates how partial order embeddings capture meaningful structures in data. Given a shadow cone embedding of mammal taxonomy, we are interested in predicting the order of a species. For example, the order of American beaver is rodents, and the order of Bonobo is primates. To construct the classification task, we first find all the nodes in the mammal graph that represent order names. This yields 10 categories, including ungulate, rodent, cetacean, etc. Then, we identify all the species names as the leaf nodes with no outgoing links. Lastly, we predict the category of each species using a softmax probability distribution based on distance. Specifically, we measure the distance between the cone central axis of the species node and those of the 10 category nodes. In the Poincaré-half space model, this distance is simply:

$$d(x, y) = \frac{1}{\sqrt{k}} \text{arcosh}(1 + \frac{||\mathbf{x} - \mathbf{y}||^2}{2 x_n y_n}),$$

where we $x_n$ and $y_n$ are both fixed to be 1. Therefore, this distance could also be thought of as the hyperbolic distance confined to the $x_n = 1$ plane.

We note that the shadow cone embedding optimizes for the entailment relations among nodes, and does not directly optimize this classification task and its loss. Yet a simple distance-based classification achieved an accuracy 94.80% with a 5D Umbral half-space cone and 73.23% with a 2D Umbral half-space cone.

## H.2 INTERPRETATION OF LIGHT SOURCE RADIUS

We modified our existing framework to learn light source radius from data. Fig 11 and 12 show the learned embedding as well as light source radius. For the mammal dataset, the source radius quickly expands from its initial size, resulting in a converged radius that is very large compared to the distribution of learned embeddings. We speculate that this might be a result of the connectivity of the underlying data: the mammal graph is highly interconnected, featuring only four disconnected components. A large light source can cast large penumbral shadows, making it easier to model highly connected DAGs. We would be interested in exploring whether different data can induce radius of different sizes, and quantify if there is a relation between source radius and graph connectivity.

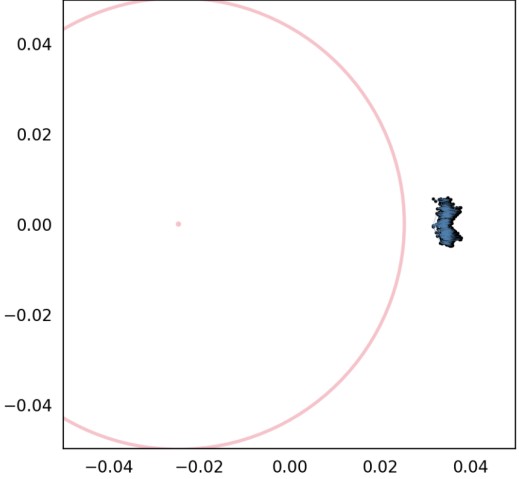

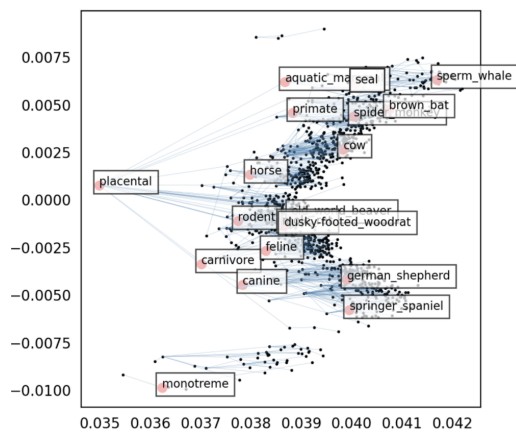

Figure 11: A global view of learned Mammal subgraph embeddings with a trainable light source radius, using Penumbral-Poincaré-ball.

Figure 12: A zoomed-in view of learned Mammal subgraph embeddings with a trainable light source radius, using Penumbral-Poincaré-ball.

## I    ANTUMBRAL CONES

For completion, we discuss the third and final category of celestial shadows - the antumbral shadows. Antumbral shadows occur under two necessary conditions: 1. The radius $r$ of the object must be smaller than the radius $R$ of the light source. 2. At least a portion of the object must be located outside the light source. In Figure 13, we illustrate antumbral cone in the half-space setting, where shadows are generally not axially symmetric.

Let $l$ be geodesics tangent to the surface of the light source $\partial\mathcal{S}$ and the object $\partial u$, such that $u$ is between $\partial\mathcal{S}$ and the intersection $u'$ of light paths. The antumbral shadow of $u$ is then defined as the penumbral shadow of $u'$. Note that by construction, for any object $u$ with well-defined antumbal cone, it is always possible to find a surrogate point $u'$, whose penumbral shadow is identical to the antumbral shadow of $u$.

Therefore, to encode relation $u \preceq v$ using antumbral shadows, it is equivalent to require their surrogate points to satisfy $u' \preceq v'$ in the penumbral cone formulation. This establishes an equivalence between the entailment relations of antumbral and penumbral formulations.

Figure 13: Antumbral Cone in Half-space

## J    FUTURE WORKS

**Geodesic Convexity of Penumbral Cones.**    In our experiment thus far, penumbral cones have not demonstrated performance on par with umbral cones in the half-space model, potentially due to their height limit and variable aperture. However, we would like to highlight a theoretical advantage unique to penumbral cones - geodesic convexity. Suppose $v_1$ and $v_2$ are both within the penumbral cone of $u$, then the entire geodesic segment $\overline{v_1 v_2}$ also resides within the penumbral cone. Such convexity lends itself to more meaningful geometric operations, such as interpolation. We thus conjecture that penumbral embeddings are better suited for word2vec or GloVe-style semantic analysis (Tifrea et al., 2018). Semantic analysis in Euclidean spaces necessitates the comparison of vectors in Euclidean space, which is straightforward. However, comparing geodesics in general Riemannian manifolds requires careful approaches using methods such as exponential maps and parallel transport, which we defer to future works.

**Downstream Tasks: Beyond the F1 Score.**    While all experiments in this study are evaluated in terms of classification scores, the potential of hierarchy-aware embedding extends beyond entailment

relation classifications. As pointed out in (Tseng et al., 2023), one can substantially enhance the performance of various attention networks by substituting their kernels with shadow-cone-based kernels that explicitly take into account the hierarchical relationships among data. In this vein, we are keen on exploring other downstream applications that utilize hierarchy-aware embedding beyond classification tasks, such as media generation (images, text, or sound) within or guided by hyperbolic space embeddings.

**Multi-relation Embeddings.** If the objective is to develop a meaningful embedding for downstream tasks, rather than solely focusing on entailment classification for a single type of relation, then it is reasonable to enrich the same embedding simultaneously with various types of relations, such as entailment and causality. This can be done while relaxing the classification accuracy for each relation types. Our framework readily facilitates this, as we can utilize multiple light sources, each casting shadows of a distinct color that captures a different type of relation. An example is depicted in Fig 14, which is set in Euclidean space for simplicity.

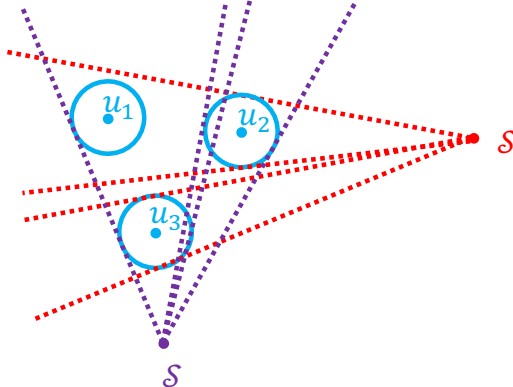

Figure 14: Multi-relation Embedding in Euclidean Space. Marked in purple and red are the two different light sources, and the respective shadow boundaries they cast.

