# OpenReview forum: "Shadow Cones: A Generalized Framework for Partial Order Embeddings"
_ICLR.cc/2024/Conference — ICLR 2024 poster_

### Official Review · Reviewer_irmp · 2023-10-22

**Soundness:** 4 excellent
**Presentation:** 2 fair
**Contribution:** 3 good
**Rating:** 6
**Confidence:** 4

**Summary:**

The author proposed physically-intuitive embedding method for partial order hierarchy. The method uses shadow cones, which generalize existing hyperbolic entailment cones. The author also constructed the algorithm to optimize the shadow cones.

**Strengths:**

1. The paper provides novel multiple variants of the existing hyperbolic entailment cone method. The techniques provided for handling shadow cones are significant technical contributions in the area.
1. The physics-inspired explanation, with plenty of intuitive figures, of the proposed method helps readers understand the proposed method's concept and algorithm. The explanation also provides another explanation of why existing hyperbolic entailment cones cannot be defined for a point around the center of the Poincare ball.
1. Table 1 summarizes the proposed 4 methods well.

**Weaknesses:**

Overall, the paper is well-written as a technical report but has room for improvement in terms of scientific discussion or presentation.
1. Although the light-source-shadow-based explanation is quite helpful in understanding WHAT the proposed methods are doing, but does not explain at all WHY they do so. Specifically, we do not see what the shape and size of the light source and the size of embedding in the umbral cases imply in the context of partial order or semantics. Also, readers do not see how we select one of the 4 proposed methods depending on the situation.
1. Overall, the paper is written as a technical extension of the hyperbolic entailment cone method, but the current draft does not position the proposed method well in the context of the whole partial order embedding area. It is not an unacceptable way, but limits the readers. The authors might want to compare it with other methods such as Order embedding, Gaussian embedding, Box embedding, etc.
1. The title of the paper lacks essential keywords and confuses readers. It should include the word "shadow cone." Also, if possible, the words "hierarchy" or "partial order" could be included. Readers might think the phrase "dark side" indicates the surface of a hemisphere, rather than a shadow cone. Also, from the word "moon," readers feel the impression that the embedding has a volume. However, this is not the case for the penumbral cone embedding. I am aware that we have many styles on which we title a paper, but, at least, we need to avoid confusing readers.

**Questions:**

1. Why the experimental results by Entailment Cone and Penumbral-Poincaré-ball are different while they are equivalent according to Theorem 3.2?
1. What do the shape and size of the light source and the size of embedding in the umbral cases imply in the context of partial order or semantics?
1. How do we select one of the 4 proposed methods depending on the situation?

---

> ### Author Response · Authors · 2023-11-21
> **Author response**
>
> **What do the shape and size of the light source and the size of embedding in the umbral cases imply in the context of partial order or semantics?**
>
> This is a very interesting question that we would like to explore with an additional experiment. Specifically, we modified our existing framework to learn source radius from data. **Figure 3 and 4 of the supplementary rebuttal pdf** show the learned embedding as well as light source radius. For the mammal dataset, the source radius quickly expands from its initial size, resulting in a converged radius that is very large compared to the distribution of learned embeddings. We think this might be a result of the connectivity of the underlying data. The mammal graph has only four disconnected components, and is thus highly interconnected. A large light source can cast large penumbral shadows, making it easier to model highly connected DAGs. In follow-up work, we would be interested in exploring whether different data can induce radius of different sizes, and quantify if there is a relation between source radius and graph connectivity.
>
> **How to select one of the 4 proposed methods depending on the situation?**
>
> In general, we recommend using shadow cones in the half-space model as there’s no $\epsilon$-hole issue compared to the Poincaré-ball model. If the goal is to represent partial order data with maximal accuracy as represented by the F1 score, then our experiments show Umbral-half-space to be the best method.
>
> If there are requirements or potential benefits from geodesic convexity, we recommend using the Penumbral-half-space cone. For example, if one were to perform interpolations between nodes, then perhaps Penumbral-half-space is superior as it achieves a relatively high F1 score, and its cones are geodesically convex.
>
> For multi-relation embeddings, where one desires to learn the sizes and positions of multiple light sources, each casting a different colored shadow, we recommend the Penumbral or Umbral-Poincaré-ball, because the half-space formulations restrict the size and position of the light source. In umbral-half-space, the light source is placed at infinity, while in penumbral-half-space, the light source is infinitely large. We discussed multi-relation embeddings in Appendix H.
>
> We sincerely thank the reviewer for the thoughtful suggestions on a more representative title. We agree and plan to change our title in the updated manuscript to: **Shadow Cones: Unveiling Partial Orders in Hyperbolic Space**. We would like to refer the reviewer to our general response, where we explain the performance disparity between Penumbral-Poincaré-ball and entailment cones.

---

> > ### Comment · Reviewer_irmp · 2023-11-22
> >
> > Thank you for your rebuttal comments.
> >
> > - *This is a very interesting question...*
> >
> > **My comments**: Thank you for your comments. I feel the results are very suggestive. It partially answers my question. However, what I exactly wanted to ask is your motivation to have a non-zero size embedding and light source, relating it to partial order or semantics. As I pointed out, not explaining at all WHY you proposed each of the four methods is a significant concern to me.
> >
> > - *If the goal is to represent partial order data with maximal accuracy as represented by the F1 score, then our experiments show Umbral-half-space to be the best method.*
> >
> > **My comments**: I appreciate it. However, it sounds an ad-hoc reason in the sense that we first need to see the experimental results to understand the advantage of each method. It would be ideal to have a non ad-hoc explanation. Specifically, I would request the authors to explain the motivation for proposing each of the four proposed methods before introducing it in the paper so that readers can understand the difference in the motivations of the four proposed methods.
> >
> > - *We sincerely thank the reviewer for the thoughtful suggestions*
> >
> > **My comments**: I'm glad if I contributed to the clarity of the title. At the same time, I would like the authors to double-check whether my suggestion matches your paper's motivation the most. I believe the phrase "shadow cone" should be in the title, but if you simply say "Unveiling Partial Orders," readers would immediately think the motivation had already been achieved by (Ganea et al. 2018) and it might lose readers. I am aware that I might have given you pressure on that matter. I am also aware that, possibly, you were unwilling to add the phrase "partial order" but had to add it since I encouraged so as a reviewer. I apologize for it if it was the case. In any case, as a member of the ICLR community, I simply hope that you will have a title that matches your paper the best. I hereby declare that I do not object to the paper title as long as the authors include the phrase "shadow cone." This is to make sure the author will not feel pressure for an unnecessary change of the title.
> >
> >
> > **Overall comments**: I will keep the current score. The reason I could not raise my score simply lies in the presentation of the current manuscript. I believe that the presentation should be improved more but at the same time the technical parts are written well. I feel the benefits the ICLR community receives from the immediate acceptance of this paper surpass the benefits we receive from further presentation revision. This is why this paper is above the acceptance threshold. Still, I strongly encourage the authors to continue improving the presentation in the camera ready so that readers can understand the difference in the motivations of the four proposed methods

---

> > > ### Author Response · Authors · 2023-11-23
> > > **Author Response 2**
> > >
> > > Thank you for the comments!!
> > >
> > >
> > > **Response to first 2 comments:**
> > >
> > > We would like to remind the reviewer that all of our cone formulations are motivated by the same overarching idea: $u \preceq v$ if and only if the shadow cast by $v$ is a subset of the shadow cast by $u$. We draw inspiration from the relationship between sun-moon-earth. In physics, there are several different categories of shadows: [umbral, penumbral and antumbral shadows](https://en.wikipedia.org/wiki/Umbra,_penumbra_and_antumbra). Umbral shadow arises when the light source is completely blocked by the occluding body, and penumbral shadow arises when the light source is partially obscured by the occluding body.
> > >
> > > We classify shadow cones according to the category of their shadows, i.e., umbral cones for umbral shadows and penumbral cones for penumbral shadows. To answer your first comment, the shape of the light source and the object embedding is dictated by the physicality of different types of shadows, and is not an arbitrary choice. Umbral shadow, which results from total eclipse of a point light source, can only be produced by both point-like light sources and objects with volume. Penumbral shadow, on the other hand, which results from partial eclipse of a light source with volume, is produced by a point object. When both the light source and objects have volume, antumbral shadow is produced, which we show in Appendix G to be mathematically equivalent to penumbral shadow formulations.
> > >
> > > We introduce shadow cones not only in the commonly adopted Poincaré-ball model, but also in the Poincaré half-space model to solve the $\epsilon$-hole issue, which gives us in total 4 (=2$\times$2) proposed cone formulations. We hope this clarifies our motivation for proposing each of the four proposed methods.
> > >
> > > In addition, we have a different view regarding the experimental results. Specifically, as summarized in Table 1 of the current manuscript, different formulations have different properties, such as geodesic convexity, light source positions and size. We cannot conclude that one cone formulation is always better than the rest. The optimal choice depends on certain types of data or task objectives.
> > >
> > > **Comment related to title:**
> > >
> > > We thank the reviewer again for the considerate and thoughtful comments. We eventually decided to adopt the following title: **Shadow Cones: A Generalized Framework for Partial Order Embedding**
> > >
> > > Lastly, we will update the current manuscript shortly and keep improving the presentation in the camera ready, if granted the chance.

---

### Official Review · Reviewer_JBVb · 2023-10-24

**Soundness:** 3 good
**Presentation:** 3 good
**Contribution:** 3 good
**Rating:** 8
**Confidence:** 3

**Summary:**

This paper considers the problem of learning representations to model hierarchical relationships using hyperbolic space. The authors introduce several formulations of *shadow cones*, a novel and unifying framework of physics inspired representations. Entailment relations are defined by the containment of shadows of the object representations given a fixed light source and boundary if hyperbolic space. The authors present two different shadow formulations each in two different hyperbolic spaces. A smooth loss function is introduced to train shadow cones in a stable manner, avoiding some of the training issues presented in prior work. The effectiveness of the presented method is shown on standard open source datasets.

**Strengths:**

* The proposed approach is novel and well-motivated, avoiding many of the issues in prior work
* The proposed representation formulation encapsulates previous approaches
* The paper is well written and includes many intuitive figures and diagrams to illustrate the proposed approach
* The empirical results demonstrate the efficacy of the proposed methodology on standard datasets for this task

**Weaknesses:**

* The paper uses quite a bit of space introducing all four formulations of shadow cones, leaving less space for additional empirical results
* At the end of Section 4, the authors make claims about ease of training, but do not justify claims beyond just a statement by fiat.
* The proposed approaches are only compared against a single baseline -- even though this is the supposed state of the art in hyperbolic representations it would be beneficial to see comparisons with other methods.
* The empirical results are all demonstrated on the same singular type of task. It would benefit the paper to see experiments on different types of tasks such as collaborative filtering.

**Questions:**

* What advantages might this method have over a different representation learning paradigm such as box embeddings (e.g. [1])?
* Why is there such a big discrepancy in the results between Penumbral-Poincaré-ball and Entailment cones in Table 2? These formulations are mathematically equivalent, but perform quite differently in some cases. Is this attributable to the loss function or training procedure in some way?
* What is the relationship between convexity and performance? The non-convex method seems to perform better. Why is this the case?
* What are the potential challenges to learning representations in the half-space formulation?

[1] Dasgupta S, Boratko M, Zhang D, Vilnis L, Li X, McCallum A. Improving local identifiability in probabilistic box embeddings. Advances in Neural Information Processing Systems, 2020

---

> ### Author Response · Authors · 2023-11-21
> **Author response**
>
> **Claim on the Ease of Training**
>
> Our training routine is less complicated than that of the Entailment Cone, in the following two ways.
> 1) While the entailment cone uses pretrained 100-epoch embedding as initialization in order to avoid the $\epsilon$-hole issue in the Poincaré-ball, we simply use random initialization around the origin as a one-stage approach.
> 2) Our half-space based cones and embeddings have no $\epsilon$-hole issue at the origin, and need no extra steps to project the “stray” points out of the hole.
>
> **Relationship between convexity and performance**
>
> This is an interesting question. As discussed in Appendix H, future works, we don’t have a clear answer to it now. We expect geodesic convexity to be beneficial for certain types of data and the task objectives, which we would like to explore in follow-up works.
>
> **Challenges to learning representation in the half-space formulation**
> There are always precision-related challenges in learning representations in any models of hyperbolic space. Such problems occur when the embedding is near the boundary of the model, i.e., $||x|| = 1$ in the Poincaré-ball model and $x_n=0$ in the Poincaré-half-space model. The limited precision of underlying floating-point computations (imprecision issue) can lead to unbounded error. Yet, the half-space model is already less prone to this issue compared to other models of hyperbolic space like the Poincaré-ball and Lorentz model, as studied in some previous work.
>
> We would like to refer the reviewer to our general response for further comparisons with the box embedding baseline, the reason for the disparity between Penumbral-Poincaré-ball and entailment cones and other downstream tasks.

---

> > ### Comment · Reviewer_JBVb · 2023-11-21
> >
> > Thank you for your response. All of my questions have been adequately answered. I'm happy to keep my score as it stands.

---

### Official Review · Reviewer_s6e6 · 2023-10-28

**Soundness:** 2 fair
**Presentation:** 3 good
**Contribution:** 2 fair
**Rating:** 6
**Confidence:** 3

**Summary:**

Hyperbolic space have been shown to be particularly well suited to encode the latent structure of complex objects such as trees or graphs. In particular, using random graphs with an hyperbolic latent space are known to exhibit properties often found in real world networks such a the small world phenomenon or the scale free property.

In this paper, the authors introduce the so-called shadow cones framework: a methodology that allows to define a partial order embedding of hierarchical data. The authors show that such framework can be particularly efficient to learn a latent embeddings of nodes for trees or more generally for DAGs. Their method extends to the concept of entailment cones previously introduced by Ganea. On three different datasets, the authors show that using their method to learn the hierarchical structure of partially observed DAGs can allow to infer unseen edges better than previous approaches.

**Strengths:**

- Hyperbolic spaces have been shown to be very promising latent space to model graph data. This paper is focused on DAGs and proposes a new framework to learn embedding for hierarchical data.

- The authors did an excellent job in ensuring the comprehensibility of their method, notably by incorporating highly insightful illustrations.

- As a by product, the paper provides a comprehensive overview of various approaches to incorporate 'shadows' for embedding points in hyperbolic space, employing the concepts of umbral and penumbral cones. This lucid exposition elucidates the source of the limitations observed in prior methods, particularly with regard to the $\epsilon$ hole problem.

**Weaknesses:**

- I think the authors should better stress the concrete applications of their method. For example, are there applications related to structure learning?

- In my opinion, the authors should include in their comparison methods that do not rely on hyperbolic spaces (but rather on Euclidean ones for example or with manifold with positive curvature).

**Questions:**

I thank the authors for their nice submission.

Apart for my questions presented in the previous section (i.e. use of the method for applications and comparison with methods not using hyperbolic spaces), I would be interested to know if the method could be used in more general settings. In particular:

- How the method could be used for real world graphs that are not (exactly) DAGs?

- How the method could be used for multiclass hierarchical problems?

Here are some typos and additional comments:

- In the loss function (cf. Eq(1)), if think N and P are not properly defined in the text.

- At the end of the first paragraph of the introduction of Section 3, I think there is an error. Should the last sentence rather be "Specifically, $v$ is in the shadow cone of $u$ iff $v \subset$ the shadow of $u$." ?

- After Theorem 4.2, I think the first "their" should be removed from the sentence "which is
designed to draw child nodes v closer to their the cones of their patent nodes".

---

> ### Author Response · Authors · 2023-11-21
> **Author response**
>
> **How the method could be used for real world graphs that are not (exactly) DAGs?**
>
> We propose shadow cones as a novel partial order embedding framework in hyperbolic space. Thus, we experiment on datasets with partial orders. Graphs that are not (exactly) DAGs deviate from the field of partial order embedding. For example, knowledge graphs, edge relations in KGs contain symmetric and other more complicated relations, which are not partial orders. Naively encoding them is likely to fail. It’s an interesting and growing area to embed general (not necessarily DAG) graphs within a partial order embedding framework. An example is embedding with minimal spanning trees. Such methods are beyond the scope of this paper.
>
> **How the method could be used for multiclass hierarchical problems?**
>
> We assume the reviewer is referring to multi-relation embeddings. This can be achieved using multiple light sources, with each light source casting shadows of one color that captures one type of relation. We briefly illustrated this idea in Appendix H. Multi-relation embedding is an exciting direction that we are interested in exploring in follow-up works.
>
> **Comparison with methods that do not rely on hyperbolic spaces**
>
> We would like to refer the reviewer to our general response, where we compare our proposed shadow cones against box-embeddings, which is a type of Euclidean embedding.
>
> We thank the reviewer for pointing out the missing definition of $P$ (the set of positive relations) and $N$ (the set of negative relations), the error in section 3 paragraph 1, and after theorem 4.2. We will make sure to fix them in the updated manuscript.

---

> > ### Comment · Reviewer_s6e6 · 2023-11-22
> >
> > I thank the authors for their response to my initial review of the manuscript. I appreciate the time and effort invested in addressing the concerns raised during the review process.
> > Based on the improvements made and the addressed concerns, I have decided to maintain my current rating for the manuscript.

---

### Official Review · Reviewer_SzVy · 2023-10-29

**Soundness:** 2 fair
**Presentation:** 3 good
**Contribution:** 2 fair
**Rating:** 6
**Confidence:** 3

**Summary:**

This paper proposes the use of (light) cones to learn embeddings of posets in hyperbolic space. Specifically, the paper looks at two models depending whether the data or the light source has mass. The paper proves a variety of consistency results such the fact that inclusion of the cones form a poset.

**Strengths:**

The idea of using cones is quite interesting to me. The paper shows that it results in a loss at least at the optimal point that does preserve the partial order structure. Further, it does so in a way that gives coordinates from which we can extract the partial order. Hence I think this is quite interesting and quite novel. I think of these are strong strengths.

The paper is mostly easy to follow however there are a few details I would like added. Please see weaknesses and questions

**Weaknesses:**

1) The first weakness for me is the context for the work. I think some more discussion to related concepts such as DAG learning, which is learning DAGs from data [1,2,3], other hyperbolic hierarchical learning like hyperbolic tree learning, which is about learning trees to represent hyperbolic data or embed in hyperbolic apace [4,5], graph embeddings in hyperbolic space [6,7,8], and hyperbolic link prediction [9] would be great.

2) I think the experimental setup could be further expanded upon. First, I think it should be clarified that we can do the link prediction without the embedding step (this would have a 100% accuracy) and then mention that you do the embedding step to understand the how well the embedding performs. As part of the experiment, it would be good to have a baseline for Euclidean poset embedding. However, I admit I do not know of one. Maybe on of the DAG learning papers has a baseline that could be used.

3) Building on the above it would be good to show that such embeddings can be used other non-trivial down stream tasks and that such embeddings provide an advantage.

[1] X. Zheng, B. Aragam, P.K. Ravikumar, and E.P. Xing. Dags with no tears: Continuous optimization for structure learning. In Advances in Neural Information Processing Systems 31, 2018.\
[2] Yu, Y., Gao, T., Yin, N., & Ji, Q. (2021, July). DAGs with no curl: An efficient DAG structure learning approach. In International Conference on Machine Learning (pp. 12156-12166). PMLR.\
[3] Lachapelle, S., Brouillard, P., Deleu, T., & Lacoste-Julien, S. (2019). Gradient-based neural dag learning. arXiv preprint arXiv:1906.02226.

[4] Ittai Abraham, Mahesh Balakrishnan, Fabian Kuhn, Dahlia Malkhi, Venugopalan Ramasub- ramanian, and Kunal Talwar. Reconstructing Approximate Tree Metrics. In Proceedings of the Twenty-sixth Annual ACM Symposium on Principles of Distributed Computing, PODC ’07, pages 43–52, New York, NY, USA, 2007. ACM.\
[5] Sonthalia, R., & Gilbert, A. (2020). Tree! i am no tree! i am a low dimensional hyperbolic embedding. Advances in Neural Information Processing Systems, 33, 845-856.

[6] Chamberlain, B. P., Clough, J., & Deisenroth, M. P. (2017). Neural embeddings of graphs in hyperbolic space. arXiv preprint arXiv:1705.10359.\
[7] T. Blasius, T. Friedrich, A. Krohmer, and S. Laue. Efficient Embedding of Scale-Free Graphs in the Hyperbolic Plane. IEEE/ACM Transactions on Networking, 26(2):920–933, April 2018.\
[8] Kevin Verbeek and Subhash Suri. Metric Embedding, Hyperbolic Space, and Social Networks. Computational Geometry, 59:1 – 12, 2016.

[9] Zhe Pan and Peng Wang. 2021. Hyperbolic Hierarchy-Aware Knowledge Graph Embedding for Link Prediction. In Findings of the Association for Computational Linguistics: EMNLP 2021, pages 2941–2948, Punta Cana, Dominican Republic. Association for Computational Linguistics.

**Questions:**

I have a few questions.

1) In the loss function $P$ and $N$ haven't been defined.

2) For the cones I do not understand the boundary computation. Could the authors please expand on that? I did not see anything in the appendix either.

3) Why do we need the ball around $y$ to be in the shadow for $y$ to be in the umbral cone, but only need the point $y$ to be in the shadows for the penumbral cone? I see that the umbral method seems to perform the best. I imagine based on how the loss in formulated this pushes the point to the interior of the cone rather than leaving it on the boundary where we might have numerical issues.

4) Could the authors provide intuition for why umbral cones are not geodetically convex? I believe in this regard calling them cones might be a bit confusing, because cones in Euclidean space are not only convex, but contains these geodesic rays. I would have thought the same is true for the hyperbolic ones. Hence was surprised when I saw this result.

---

> ### Author Response · Authors · 2023-11-21
> **Author response**
>
> **Broader Contexts:**
> We sincerely thank the reviewer for kindly suggesting broader literatures to contextualize our work within. We will incorporate discussions and comparisons with the suggested papers in section 2 of our updated manuscript.
>
> **Non-trivial downstream tasks:**
> We would like to refer the reviewer to our general response section, where we discussed how cone-based attention could be used to improve the performance of attention models, and offer a simple classification task demonstrating how our partial order embedding captures meaningful structure in the mammal taxonomy data.
>
> **Question 3:**
>
> First we would like to remind the reviewer that all of our cone formulations are motivated by the same overarching idea: $u \preceq v$ if and only if the (penumbral or umbral) shadow cast by $v$ is a subset of the shadow cast by $u$. The shape of the object (single point, or ball with volume) is dictated by the physicality of different types of shadows, and is not an arbitrary choice. Penumbral shadow, which results from partial eclipse of a light source with volume, is produced by a point object.
>
> On the other hand, Umbral shadow, which results from total eclipse of a point light source, is produced by an object with volume. In order to satisfy the definition of the subset relation of (umbral and penumbral) shadows, we need the ball around objects $y$ to be in the shadow in the umbral cone, but only need the point $y$ to be in the shadow for the penumbral cone.
>
> The Umbral-half-space cone performs the best in general. The reason can be multi-folds, including loss function and optimization process. However, we attribute its main advantage to the infinite-height light source, and lack of $\epsilon$-hole at the origin.
>
> **Boundary computation and intuitive explanations why umbral cones are not geodesically convex**
>
> As illustrated in the last question, Penumbral shadow, which results from partial eclipse of a light source with volume, is produced by a point object. Therefore, the boundaries of penumbral shadow are geodesics passing through $u$ while being tangent to the light source. Note that the $\text{Penumbral Shadow}(v) \subset \text{Penumbral Shadow}(u)$ for any $v\in \text{Penumbral Shadow}(u)$, therefore, all possible children of $u$, i.e., penumbral cone of $u$ coincides with the penumbral shadow of $u$, with geodesics boundary described above.
>
> On the other hand, Umbral shadow, which results from total eclipse of a point light source, is produced by an object with volume (a ball around $u$). Therefore, the boundary of umbral shadow is geodesics passing through the point light source, while being tangent to the ball around $u$. Now, let’s look at all possible children $v$ of $u$, i.e., umbral cone of $u$. Since the objects $u$ and $v$ have volumes, then for $u \preceq v$ to be true, it must be that $v$ and its ball are entirely immersed in $u$’s shadow. Otherwise, the portion of $v$’s ball outside of $u$’s shadow can cast shadows outside. In **Figure 1 of the supplementary rebuttal pdf**, we sketch a case where $v$’s center is in $u$’s shadow, but nevertheless $u \npreceq v$, because part of $v$’s ball is outside.
>
> Hence, the boundary of umbral cones are the set of points equidistant from the umbral shadow, aka, hypercycles in hyperbolic space, a well-studied geometric object. In Euclidean space, the set of points equidistant from a straight line forms another, parallel straight line. In contrast, in hyperbolic space, the set of points equidistant from a geodesic do *not* necessarily form a geodesic, and is instead referred to as a hypercycle. In **Figure 2 of the supplementary rebuttal pdf**, we illustrate the non-convexity of umbral cones in Poincaré half-space. In green are the umbral cone boundaries, while in black are its convex hull - the minimum convex set - that contains the umbral cone. The boundaries of the convex hull are geodesics. We can see that the umbral cone is a strict subset of its convex hull, which means the umbral cone is not geodesically convex. We will include this illustration in the Appendix.
>
> We thank the reviewer for pointing out the missing definition of $P$ (the set of positive relations) and $N$ (the set of negative relations). We also agree with the suggestion on further explanation of the experimental setup. We will make sure to clarify and explain them in the updated manuscript. We would also like to refer the reviewer to our general response section, where we compare our proposed shadow cones against box-embeddings, which is a type of Euclidean embedding.

---

### Official Review · Reviewer_Arvf · 2023-10-31

**Soundness:** 3 good
**Presentation:** 3 good
**Contribution:** 3 good
**Rating:** 6
**Confidence:** 4

**Summary:**

This paper proposes a physically intuitive partial order embedding framework "shadow cones" that generalizes the well-known "hyperbolic entailment cones". Also, this framework generalizes to two different hyperbolic models, the Poincaré disk and Poincaré half-space. The experiments shows that the generalized "shadow cones" outperform the "hyperbolic entailment cones" baseline.

**Strengths:**

1. The paper is well-motivated and has a good connection to existing research.
2. It considers two hyperbolic models, the Poincaré disk and the Poincaré half-space.
3. The theoretical analysis provides nice properties of the proposed models.

**Weaknesses:**

1. Compared with the novel and interesting motivation, the description, especially the figures, lacks explanation, almost all the symbols in the figures are not mentioned in the captions or corresponding text.
2. The experiment is not very supportive of the claim. For example, In the experiment, Poincaré ball based cones performed worse than Poincaré half-space based cones. While in the important baseline "hyperbolic entailment cones", the Poincaré ball is used, if the proposed cone generalizes "hyperbolic entailment cone", it should be as good as the "hyperbolic entailment cone".
3. The use of different definitions for light source in the Penumbral cone and the Umbral cone is somewhat inconsistent.

**Questions:**

Please see the weaknesses.

---

> ### Author Response · Authors · 2023-11-21
> **Author response**
>
> **Weakness 1:**
> We kindly thank the reviewer for the suggestions on the descriptions. We will update the manuscript to explain all symbols in the figures.
>
> **Weakness 2:**
> We believe the reviewer misunderstood our claim: shadow cones generalize hyperbolic entailment cones (Poincaré-ball based) to a broad class of formulations. The hyperbolic entailment cones are mathematically equivalent to a special case of shadow cones, namely, Penumbral-Poincaré-Ball. We understand that the reviewer might be puzzled by the performance difference of hyperbolic entailment cones and Penumbral-Poincaré-Ball, given that the two are mathematically equivalent. We would like to refer the reviewer to our explanation in the general responses.
>
> Poincaré half-space based cones are another category of cones within the shadow cones, which are distinct from the Poincaré-Ball based formulations. We analyzed and discussed why Poincaré half-space based formulations perform better than Poincaré-Ball based formulations in section 5, page 9, [Discussion on the impact of $\epsilon$-hole].
>
> **Weakness 3: Different light sources in Penumbral and Umbral cones**
>
> We wanted to emphasize that whether the light source has volume is dictated by the physicality of different types of shadows, and is not an arbitrary choice. Penumbral shadow results from partial eclipse of the light source, and could only be produced by light source with volume. Umbral shadow, on the other hand, results from total eclipse of a point light source, and can be produced by both point-like light sources and objects with volume. When both the light source and objects have volume, a third type of shadow is produced - antumbral shadow, which we show in Appendix G to be mathematically equivalent to penumbral shadow formulations.

---

### Official Review · Reviewer_6mLv · 2023-11-03

**Soundness:** 3 good
**Presentation:** 2 fair
**Contribution:** 3 good
**Rating:** 6
**Confidence:** 3

**Summary:**

The authors introduce the "shadow cones" framework for constructing entailment cones in hyperbolic space. Unlike prior work that utilized nested cones in the Poincaré ball, shadow cones model partial orders based on subset relations between shadows created by a light source and opaque objects.  This framework extends beyond the Poincaré ball, allowing for more diverse formulations and hyperbolic space models. Shadow cones offer advantages over existing constructions, particularly in terms of optimization properties. Experimental results demonstrate the consistent and significant outperformance of shadow cones compared to existing entailment cone constructions across datasets of varying sizes and hierarchical structures.

**Strengths:**

1. The introduction of shadow cones presents an innovative approach to defining entailment relations in hyperbolic space. Drawing inspiration from physical phenomena adds an intuitive and captivating aspect to the concept.

2.  The authors provide a comprehensive mathematical formulation for shadow cones, enhancing the rigor of their work and enabling further exploration and development by other researchers in the field.

3.  The experimental results across four datasets demonstrate the superior performance of the proposal compared to the current state-of-the-art methods.

**Weaknesses:**

1. While the authors tested their framework on four datasets, it would be beneficial to see how the framework performs on a wider variety of datasets, including those from different domains or those with different characteristics (like KG, recommender system datasets and so on). This would provide a more comprehensive evaluation of the framework's performance and versatility.

2.The work builds on the entailment cone but a few partial order methods have been proposed and a thorough comparison or discussion is needed.

3. Captions for the figures are brief and ambiguous, hindering their readability.


[1]Capacity and Bias of Learned Geometric Embeddings for Directed Graphs.
[2]Modeling Transitivity and Cyclicity in Directed Graphs via Binary Code Box Embeddings

**Questions:**

see weakness

---

> ### Author Response · Authors · 2023-11-21
> **Author response**
>
> **Experiments on a wider variety of datasets from different domains or those with different characteristics (like KG etc.)**
>
> We propose shadow cones as a novel partial order embedding framework in hyperbolic space. Thus, we experiment on datasets with partial orders. We augment our experiments with MCG and Hearst dataset beyond the standard Wordnet benchmark. Datasets from different domains or those with different characteristics are likely to deviate from the partial order embedding area. For example, knowledge graphs, edge relations in KGs contain symmetric and other more complicated relations, which are not partial orders. Naively encoding them is likely to fail. In fact, it’s an interesting and growing area of embedding general relations in a partial order embedding framework, for example, with minimal spanning tree. Hence, we only consider datasets with partial order relations and it’s beyond the scope of this paper to experiment on datasets and tasks from other domains.
>
> We kindly thank the reviewer for suggesting further comparisons. We would like to refer the reviewer to our general response section, where we compare our proposed shadow cones against box-embeddings. We will update the captions of figures to improve the readability.

---

> > ### Comment · Reviewer_6mLv · 2023-11-21
> > **Thanks for the feedback**
> >
> > Thanks to the authors for the reply. My concerns have been most addressed. The idea is interesting and experimental results are promising.  I would like to suggest the authors revise and reupload the manuscript incorporating the responses and improving the readability as well.  I will be happy to raise my score.

---

> > > ### Author Response · Authors · 2023-11-23
> > > **Author response 2**
> > >
> > > Thank you for your thoughtful feedback! We will update the manuscript shortly incorporating the responses and improving the readability of figures and their captions.

---

### Author Response · Authors · 2023-11-21
**General response (part 1)**

We would like to thank the reviewers for their thoughtful reviews and suggestions. All reviewers recognized our proposed shadow cone framework as innovative and novel. Reviewers 6mLv and s6e6 appreciated mathematical rigor and physical insights associated with the shadow cone formulations. Reviewers s6e6 and irmp also highlighted the lucidity of our writing and illustrations.

We would now use this space to address several common questions and suggestions.

**All reviewers suggested making comparisons to Euclidean partial order embeddings or Box Embeddings.**

As suggested, we conduct experiments on Mammal with box embeddings [1,2] as another baseline, which is a partial order embedding framework in Euclidean space. Specifically, we used the latest GBC-box and VBC-box embeddings proposed in [1] from the official implementation https://github.com/iesl/geometric-graph-embedding/tree/dev/dongxu. To make apple-to-apple comparisons, we train box embeddings with the same hyper-parameters (epochs, batchsize, negative sampling rate etc.) and fractions of non-basic edges as shadow cones in the paper, and evaluate the generalization performance of the embedding on the same test set. The results are listed below:

|                           | dim=2   |         | dim=5   |         |
|---------------------------|---------|---------|---------|---------|
| non-basic-edge percentage | GBC-box | VBC-box | GBC-box | VBC-box |
| 0%                        | 23.4%   | 20.1%   | 35.8%   | 30.9%   |
| 10%                       | 25.0%   | 26.1%   | 60.1%   | 43.1%   |
| 25%                       | 23.7%   | 31.0%   | 66.8%   | 58.6%   |
| 50%                       | 43.1%   | 33.3%   | 83.8%   | 74.9%   |
| 90%                       | 48.2%   | 34.7%   | 97.6%   | 69.3%   |

Our shadow cone embeddings outperform box embeddings in nearly all cases, except the one case with GBC-box in dim=5 and trained with 90% non-basic edges.

**Reviewers Arvf, JBVb and irmp remarked upon the different performances of Entailment Cones and Penumbral-Poincaré-Ball, despite the two being mathematically equivalent.**

The performance difference results from three sources: initialization, energy function, and loss function.
As noted in section 5, the Entailment Cone framework uses pre-trained 100-epoch embedding from [Nickel & Kiela 2017] as initialization. However, we initialize our embedding simply with a bounded uniform distribution around the origin.
As discussed in section 4, we use a distance-based energy function that measures how far a node is from the target node’s cone boundary, while the Entailment Cone uses an angle-based energy. We argued in this section how the distance-based energy offers a more nuanced understanding of hierarchical relationships.
Lastly, the Entailment Cone uses the max-margin energy loss which, when combined with their angle-based energy, suffers from vanishing gradients for misclassified negative samples in the cone. Our approach has no such issues, and it uses a contrastive loss.

[1] Modeling Transitivity and Cyclicity in Directed Graphs via Binary Code Box Embeddings

[2] Capacity and Bias of Learned Geometric Embeddings for Directed Graphs.

---

> ### Author Response · Authors · 2023-11-21
> **General response (part 2)**
>
> **Reviewers SzVy, s6e6, and JBVb expressed interest in seeing downstream tasks where our proposed shadow cone framework can prove useful.**
>
>
> One promising application is mentioned in section 6: Tseng et al. 2023 ([Coneheads: Hierarchy Aware Attention](https://arxiv.org/abs/2306.00392)), proposes a shadow-cone based hyperbolic attention mechanism, which can be used as a drop-in replacement for dot-product attention in transformers. Cone attention associates two points by the depth of their lowest common ancestor in a hierarchy defined by hyperbolic shadow cones. Tseng et al. tested cone attention on various models and tasks, and demonstrated how shadow cone-based attention could improve task-level performance over dot product attention and other baselines.
>
>
> Here, we would like to supplement with a simpler experiment that demonstrates how partial order embeddings capture meaningful structures in data. Given a shadow cone embedding of mammal taxonomy, we are interested in predicting the order of a species. For example, the order of American beaver is rodents, and the order of Bonobo is primates. To construct the classification task, we first find all the nodes in the mammal graph that represent order names. This yields 10 categories, including ungulate, rodent, cetacean, etc. Then, we identify all the species names as the leaf nodes with no outgoing links. Lastly, we predict the category of each species using a softmax probability distribution based on distance. Specifically, we measure the distance between the cone central axis of the species node and those of the 10 category nodes. In the Poincaré-half space model, this distance is simply:
>
>
> $$
> d(x,y)=\frac{1}{\sqrt{k}}\text{arcosh}(1+\frac{||\bf{x}-\bf{y}||^2}{2x_n y_n}),
> \$$
>
>
> where we $x_n$ and $y_n$ are both fixed to be 1. Therefore, this distance could also be thought of as the hyperbolic distance confined to the $x_n=1$ plane.
>
>
> We note that the shadow cone embedding optimizes for the entailment relations among nodes, and does not directly optimize this classification task and its loss. Yet a simple distance-based classification achieved an accuracy 94.8%.
>
>
> | model                | accuracy |
> | -------------------- | -------- |
> | 5D umbral half-space | 94.80%   |
> | 2D umbral half-space | 73.23%   |

---

### Author Response · Authors · 2023-11-23
**Updated manuscript**

We would like to thank all reviewers again for taking time to review our paper and provide thoughtful suggestions. We have updated the manuscript in the following aspects:


1) modify captions of all figures to illustrate all appearing symbols and improve readability, as suggested by most reviewers
2) include GBC-box and VBC-box embeddings as Euclidean baselines in experiments and related work section, as suggested by most reviewers
3) change tile to: **Shadow Cones: A Generalized Framework for Partial Order Embedding**, as a result of discussions with reviewer irmp
4) add section H in appendix for additional experiments and discussions on seeing downstream tasks and the ablation experiment with trainable light source radius, as suggested by reviewer SzVy, s6e6, JBVb and irmp
5) add section B in appendix to intuitively explain shadow cones’ boundary computations and non-geodesic-convexity of Umbral cones, as suggested by reviewer SzVy
6) Justify the claim of ease of training in section G, as suggested by reviewer JBVb
7) include missing definitions of P, N in equation 1 and fix typos, as suggested by most reviewers


– We thank reviewer SzVy for kindly suggesting broader literature. We tried to incorporate as many suggestions from all reviewers as possible without significantly changing the structure of the paper, however, with the page and time limit, we plan to discuss suggested references in a future version of the paper.

---

### Meta-Review · Area_Chair_ux3q · 2023-12-05

**Metareview:**

The paper introduces a new method, the ``shadow-cones’’ framework to better capture hierarchical structure in the hyperbolic space. The main advantage of the new method is that it admits new optimization properties that make it possible to design methods with better practical performances.

The paper presents some new interesting results in an important area so it would be a good addition to the ICLR program.

A general suggestion for the author(s) would be to make the paper more accessible by explaining the main definition in the paper more carefully(maybe also by adding figures). Also adding comparison with additional baselines will strengthen the final version.

**Justification For Why Not Higher Score:**

The paper is interesting and the presented results are nice although the experimental analysis could be strengthen

**Justification For Why Not Lower Score:**

The paper introduce some new concept that can be of general interest

---

### Decision · Program_Chairs · 2024-01-16

Accept (poster)